# Tunable anti-ambipolar vertical bilayer organic electrochemical transistor enable neuromorphic retinal pathway

Zachary Laswick[1,4], Xihu Wu[2,4], Abhijith Surendran [1], Zhongliang Zhou[2], Xudong Ji [1], Giovanni Maria Matrone [1] ✉, Wei Lin Leong [2] ✉ & Jonathan Rivnay [1,3] ✉

Increasing demand for bio-interfaced human-machine interfaces propels the development of organic neuromorphic electronics with small form factors leveraging both ionic and electronic processes. Ion-based organic electrochemical transistors (OECTs) showing anti-ambipolarity (OFF-ON-OFF states) reduce the complexity and size of bio-realistic Hodgkin-Huxley(HH) spiking circuits and logic circuits. However, limited stable anti-ambipolar organic materials prevent the design of integrated, tunable, and multifunctional neuromorphic and logic-based systems. In this work, a general approach for tuning anti-ambipolar characteristics is presented through assembly of a p-n bilayer in a vertical OECT (vOECT) architecture. The vertical OECT design reduces device footprint, while the bilayer material tuning controls the anti-ambipolarity characteristics, allowing control of the device's on and off threshold voltages, and peak position, while reducing size thereby enabling tunable threshold spiking neurons and logic gates. Combining these components, a mimic of the retinal pathway reproducing the wavelength and light intensity encoding of horizontal cells to spiking retinal ganglion cells is demonstrated. This work enables further incorporation of conformable and adaptive OECT electronics into biointegrated devices featuring sensory coding through parallel processing for diverse artificial intelligence and computing applications.

As artificial intelligence applications continue to increase and integrate into daily life, the demand for low-cost, highly efficient, bio-interfaced computing hardware stands out as a global technological challenge[1,2]. Conventional silicon electronics operate following the paradigm of von Neumann architecture which depends on several circuital elements such as transistors, inverters, and their combination into logic circuits. However, these systems show poor biological compatibility on both an interfacing level, due to the inherent rigidity of conventional inorganic, CMOS transistors, and on a computing level, since they operate through serialized computational functions, thereby limiting their speed and increasing their power consumption[2,3]. Recent advances in neuromorphic systems specifically aim to bypass this "von Neumann bottleneck" by mimicking the massively parallel and event-driven computational functions of the human brain thus offering increased speed, efficiency, and performance over conventional computing algorithms[2].

[1]Department of Biomedical Engineering, Northwestern University, Evanston, IL, USA. [2]School of Electrical and Electronic Engineering, Nanyang Technological University, 50 Nanyang Avenue, Singapore 639798, Singapore. [3]Department of Materials Science and Engineering, Northwestern University, Evanston, IL 60208, USA. [4]These authors contributed equally: Zachary Laswick, Xihu Wu. ✉e-mail: giovanni.matrone@northwestern.edu; wlleong@ntu.edu.sg; jrivnay@northwestern.edu

Organic mixed ion-electron conductors (OMIECs) are emerging for bioelectronic and neuromorphic applications due to their favorable properties, including demonstrated biocompatibility, flexibility/stretchability, and ion-based tunability[4]. Specifically, organic electrochemical transistors (OECTs) require low operational voltages which serve to modulate the conductivity of their organic mixed ion-electron conductor channel resulting in high transconductance compared to field effect transistors[3,4]. The ion-based operation of these systems mimics neuronal ion-flux communication and neurotransmitter-receptor binding, promising future interaction with biological tissues as adaptive bio-interfaces[5–8]. As a result, OECTs have also been essential to design artificial neural circuits[9–11]. Within neuromorphic systems, OECT-based biologically-inspired synapses exhibiting both long/short term plasticity and spike timing-dependent plasticity have been developed to directly interface with living tissue[5,6].

Integrate and fire models have been used to design artificial spiking networks, but these circuits require multiple OECTs in a configuration comprising two complementary inverters and a switch, thereby limiting circuit miniaturization[9]. They also cannot compete with the biological realism of Hodgkin–Huxley(HH) neuron models, which closely replicates the flow of sodium and potassium ions within a biological neuron, serving as the fundamental mechanisms to generate action potentials[12]. Such biologically realistic circuit models have been demonstrated in silico for pattern classification, artificial retinas, and neurological disease models[13,14]. The sodium channel ion flux exhibits an inherent anti-ambipolar function which is critical for generating an action potential and must be replicated in neuromorphic systems[10,12]. Indeed, anti-ambipolar transistors are ideal neuromorphic devices since they naturally exhibit a characteristic positive and negative transconductance within their OFF-ON-OFF transfer curve[10,15]. Therefore, HH neuron circuits have been developed using anti-ambipolar OECTs based on poly(benzimidazobenzophenanthroline) (BBL), one of the few materials so far reported with stable anti-ambipolar characteristics[10]. BBL exhibits inherent anti-ambipolarity due to the overfilling of band states within the organic semiconductor that inhibits electronic conduction across the material at high gate voltages[10,16]. The generation of spikes in HH neurons replicates action potentials comprising an ion-dependent operation similar to in-vivo neurons[10]. However, these systems usually display fixed neuron characteristics, such as spiking threshold and frequency, and limited device footprints due to their planar structure[10]. Being able to control these parameters is essential to design bio-realistic artificial neurons and account for the high specialization of their biological counterparts in the central nervous system where these metrics vary depending on the cell's functions[12]. For example, neurons with different action potential thresholds have been identified in the brain. These threshold variations represent a fundamental mechanism enabling neural microcircuits sharing a similar structure to perform a variety of computational tasks, as well as enabling increased robustness, synchronicity, sensitivity, and dynamic range in the whole central nervous system[17–20].

Another promising strategy to overcome the "von Neumann bottleneck" is represented by the development of electrically reconfigurable logic circuits, where anti-ambipolarity enables dynamic reshaping of the circuit's connectivity and functionality. Regularly, planar OECTs have been employed to develop inverters and logic gates targeting a plethora of applications, including wearable bio-interfaced closed-loop electronics[8,21–24]. However, the footprint of these circuits are limited not only by these devices' planar structure but also due to the device number required per logic function and the limited configurability of the circuits[21]. Thus, anti-ambipolarity allows the same transistor to be reconfigured in different logic operations depending on the use of positive and negative transconductance, enabling a bypass of the device number per logic function limitation presented by the "von Neumann bottleneck"[25–27]. However, the same

lack of tunability amongst the limited single-component anti-ambipolar OECTs hinders power consumption improvements, bio-interfacing and sensing capabilities.

Despite the lack of intrinsically anti-ambipolar inorganic materials, or organic materials for field effect transistors (OFET), OFF-ON-OFF transfer characteristics can be promoted through an in-series combination of p-type and n-type materials. In these in-series field effect transistors, the high-off resistance of each material is leveraged to prevent current flow when the gate voltage exceeds the region of overlap[15,25–28]. Therefore, the selection of p-type and n-type materials of an in series-transistor alters the region of conduction overlap thereby defining the bounds of the in series devices anti-ambipolar transfer curve. This allows circuits of further reduced complexity, thus facilitating scalable Gaussian probabilistic neural networks and neuromorphic spiking systems[15,28].

In this work, we demonstrate a bilayer vOECT with anti-ambipolar transfer characteristics by realizing an in-series connection of p-type and n-type OMIECs with properly selected threshold characteristics, vertically arranged through stacked layers. We propose a generalizable approach for the selection of these materials realizing anti-ambipolar transistors with tailored full width at half maximum (FWHM), turn on, turn off, and peak current gate voltages. Control over these transistor metrics enables scalable, low voltage, tunable OECTs which serve to build customizable logic devices and neuromorphic spiking circuits. In the context of HH-spiking circuits, the bilayer approach serves to tune the spiking threshold of the circuit depending on the characteristics of the single anti-ambipolar transistors, in parallel with the high specialization of biological computing elements. In these cases, the vOECT architecture also enables miniaturization of circuits, while maintaining scalable fabrication methods, favoring in-sensor and point-of-care applications. Inspired by the flexibility of the human brain, neuromorphic spiking systems are designed to receive diverse stimuli such as light, pressure, temperature, and perform sensory coding[1,29–32]. Although biological encoding/computing mechanisms are far from pure Boolean logic, specific biological interactions can be described with a combination of binary algebra and multiple-state logic gates[29–31]. For instance, in the retina, cones and rods synapse with horizontal cells to constitute a specific preprocessing unit for wavelength and light intensity signals-encoding whose operations correspond to the boolean logic function "AND". As such, the building blocks represented by the HH-spiking circuit and the logic gates are combined to mimic light-wavelength encoding retina-inspired functions. By exploiting the anti-ambipolar OFF-ON-OFF nature of the logic-gate units, we demonstrate preprocessing functions which activate essential firing patterns corresponding to specific light intensity and wavelength conditions[32–34]. This opens the possibility to seamlessly integrate anti-ambipolar logic circuits with spiking circuits to closely mimic specific architecture of the retinal circuitry and achieve complex preprocessing functions. Hence, the bilayer architecture formulates a novel organic electronics concept which enables fine control of the vOECTs anti- ambipolar characteristics, with applications that span the diverse fields of alternative-to-conventional electronics where the control of electrical characteristic in small-scale integrated circuits is of crucial importance.

## Results
### Anti-ambipolar bilayer vOECTs
The schematic in Fig. 1c illustrates the structure of the bilayer vOECT. This device adopts a three-terminal architecture, in which the OMIEC layers are vertically patterned between two electrodes, with an Ag/AgCl pellet serving as the gate electrode and PBS as the electrolyte. To fabricate anti-ambipolar bilayer OECTs, we utilized the commercially available enhancement-mode n-type ladder polymer, BBL, and the depletion-mode p-type poly(3,4-ethylenedioxythiophene):poly(styrene sulfonate) (PEDOT:PSS), and deposited these materials by sequential spin coating to form an in-series vertical structure (Fig. 1b).

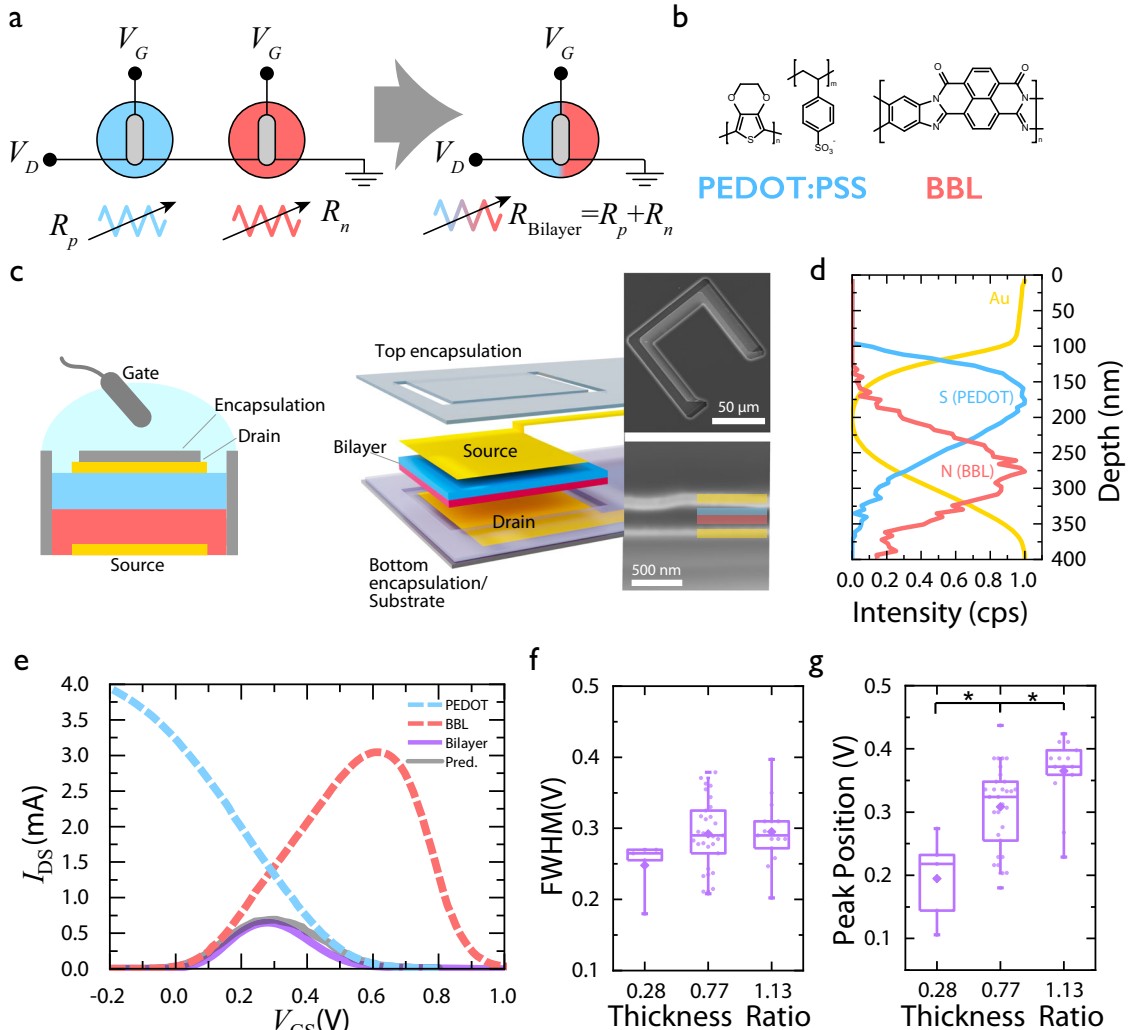

**Fig. 1 | Bilayer vOECT characterization. a** Diagram detailing the in-series connection of the bilayer materials structure and the equivalent circuit using variable resistors ($R_n$ for n-type resistance and $R_p$ for p-type resistance). **b** Materials for each layer of the bilayer are included where the n-type BBL represents the red bottom layer and the p-type PEDOT:PSS represents the blue top layer. **c** Schematic of the bilayer vOECT structure, where top and bottom electrodes are separated by the bilayer within the active area, or an insulating parylene layer outside the channel, and the top electrode is passivated using a photoresist layer. SEM image of the device is included for reference. **d** XPS data from the materials bilayer where sulfur (blue) indicates the presence of PEDOT:PSS and nitrogen (red) indicates the presence of BBL. **e** Transfer curves of the BBL-PEDOT bilayer, BBL, and PEDOT vOECTs overlaid with a prediction of the BBL-PEDOT bilayer using the circuit in (**a**). ($W = 100\ \mu m$, $V_{DS} = 0.1\ V$) (**f**) Change in full width at half maximum with differing ratios of BBL to PEDOT thickness. There is no statistically significant difference in FWHM ($n > 5$, $p = 0.12$). **g** Change in peak Position with differing ratios of BBL to PEDOT thickness. There are statistically significant differences for peak position (one-way ANOVA with Tukey post hoc test $p < 0.05$, $n > 5$, $p = 0.015$). Both panel (**f**, **g**) display mean (diamond), median (line), quartiles (box), and 5–95 whiskers.

Each OMIEC film represents a variable resistor ($R_{\text{p-type}}$ and $R_{\text{n-type}}$) in the electronic circuit of the resulting bilayer ($R_{\text{Bilayer}}$), and operates as an independent OECT (Fig. 1a). Note, the direct in-series deposition of p- and n-type materials ensures contact between the layers and also allows for alternative scalable OECT device fabrication methods (Fig. 1c, d). As such, the final structure of the bilayer device is shown in Fig. 1c SEM image, with XPS data highlighting the two distinct layers of OMIEC materials, with a slight intermixing layer[35–37] (Figs. 1d and SI Note 1). The thickness of the bottom BBL layer of the vOECT is 85 nm, while the thickness of the top PEDOT:PSS layer is 110 nm. As in conventional lateral OECTs, the electrical conductivity of an OMIEC is controlled by the exchange of ions between the electrolyte and OMIEC layer in response to a gate bias.

The transfer characteristics of a BBL and PEDOT:PSS single layer vOECTs, and BBL-PEDOT bilayer vOECT are displayed in Fig. 1e. Due to the vertical architecture of the devices, high $W/L$ values are designed, resulting in improved conductivity and transconductance compared to lateral devices, where a single channel material connects coplanar source and drain electrodes (SI Note 1 and Figs. S1–S3). Furthermore, summing the resistances of the BBL and PEDOT:PSS single layer vOECTs allows prediction of the resulting bilayer transfer characteristics, establishing a "bilayer materials selection rule" to guide device design and performance across multiple material systems (Fig. 1a, e and SI Note 1). The BBL-PEDOT bilayer vOECT clearly demonstrates the uniquely anti-ambipolar OFF-to-ON-to-OFF states with increasing $V_{GS}$, yielding a peak position at $V_P = 0.308 \pm 0.066\ V$. To verify the influence of each layer's resistance on the anti-ambipolar shape, we directly modulated the PEDOT:PSS layer thickness, while maintaining the BBL layer thickness at 85 nm (Fig. 1f, g). Due to the vertical structure of the bilayer OECT, increasing the PEDOT:PSS layer thickness corresponds to an increase of the PEDOT:PSS OECT channel length in the equivalent circuit, which results in decreased conductivity of this layer. Hence, decreasing the conductivity of PEDOT:PSS within the bilayer results in a shift of the intersection point of the individual BBL and PEDOT:PSS

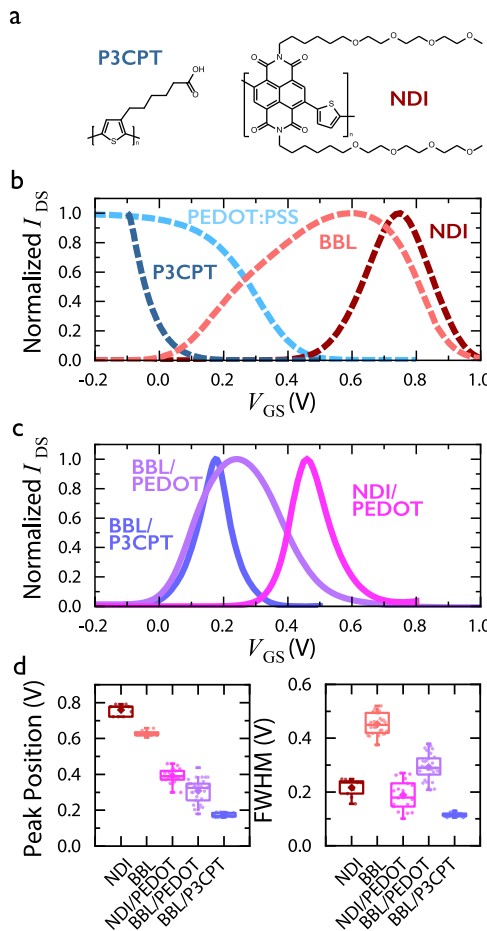

**Fig. 2 | Tuning of the anti-ambipolar characteristics by p- and n-type control.**
**a** Chemical structure of the OMIEC materials used for p- and n-type control.
**b** Normalized transfer curves of each material used in the vOECT structure
($W = 100\,\mu m$, $V_{DS} = 0.15\,V$ or $0.3\,V$ for P3CPT). **c** Normalized transfer curves of the
resulting bilayer vOECTs using the materials presented. **d** The dependence of peak
position and FWHM on the material selection ($n > 9$, mean (diamond), median
(line), quartiles (box), and 5–95 whiskers are shown, one-way ANOVA with Tukey
post hoc test $p < 0.05$ shows all are significantly different, except the FWHM of NDI
and NDI/PEDOT).

transfer curves leading to a statistically significant decrease in the peak
position of the bilayer (Fig. 1f, g). Furthermore, bilayer and individual
vOECT conductance scales with device dimensions, thereby resulting
in a minor dependence of peak position on W-value but not for FWHM
or on $V_{DS}$ (SI Note 1). As such, controlling each material's resistance
allows fine-tuning of the bilayer anti-ambipolar transfer curve's peak
position but not the full width at half maximum (FWHM) (Fig. 1f, g).
Additionally, the BBL-PEDOT bilayer vOECTs are stable for at least 750
cycles, with failure relating to degradation of the PEDOT:PSS layer
(SI Note 1).

## Material selection tuning the anti-ambipolar characteristics

Following the bilayer materials selection rule, alternative p-type and
n-type combinations are investigated for the fabrication of anti-
ambipolar transistors with desired anti-ambipolar characteristics.
The effects of material selection are primarily explored by choosing
PEDOT:PSS as a model p-type material and using two different n-type
OMEIC materials (n-type control), the ladder-polymer BBL and the
glycolated NDI n-type p(C6NDI-T)(SG303) (Fig. 2a). Due to fabrica-
tion constraints, the bilayer's bottom layer is constituted by an
n-type material, as n-type materials utilize solvents, such as acids,
that can damage an underlying p-type layer if it were cast on top (SI

Note 1 and Figs. S1–S8). Furthermore, the effect of different p-type
materials is explored by holding BBL as the n-type model material
and using two different p-type OMIEC materials (p-type control), the
conductive polyelectrolyte complex PEDOT:PSS and the conjugated
polyelectrolyte Poly [3-(5-carboxypentyl)thiophene-2,5-diyl] (P3CPT)
(Fig. 2a). Figure 2b illustrates the self-normalized transfer curves for
each material used in the vOECT structure, with BBL and p(C6NDI-T)
devices exhibiting anti-ambipolar characteristics. Figure 2c shows
both the n-type and p-type control for tuning the OECT bilayer
characteristics. The p(C6NDI-T)-PEDOT bilayer devices present a
peak position at $V_P = 0.392 \pm 0.039\,V$, which is lower than the
single p(C6NDI-T) device of $V_P = 0.759 \pm 0.031\,V$ but higher than
the BBL-PEDOT bilayer model system, due to the overlap of the
single layer characteristics (Figs. 2c, b and S6). Correspondingly,
in p-type control, the BBL-P3CPT bilayer also exhibited anti-
ambipolar characteristics, with a peak position shift to a lower vol-
tage ($V_P = 0.179 \pm 0.007\,V$), which is consistent with the prediction
based on the resistances summation (Figs. 2c, b and S6). In this
bilayer case, a minimum single layer OECTs transfer characteristic
overlap is realized, corresponding to a reduced FWHM of
$0.113 \pm 0.003$ (Fig. 2d). The NDI-PEDOT and BBL-P3CPT bilayers fol-
low a similar W-scaling (Fig. S7). On the other hand, the anti-
ambipolar characteristics of the BBL-P3CPT bilayer exhibit a slight
scaling with $V_{DS}$ due to the significant shift of the P3CPT threshold
voltage with $V_{DS}$, whereas this scaling does not apply to the NDI-
PEDOT bilayer (Fig. S7 and SI Note 1). The peak positions of these
anti-ambipolar vOECTs are statistically different ($P < 0.05$) decreas-
ing from 0.627, 0.759, 0.308, 0.392, and to 0.179 V for BBL, NDI, BBL-
PEDOT, NDI-PEDOT, and BBL-P3CPT respectively (Fig. 2d). Similarly,
the selection of materials with different regions of transfer curve
overlaps also affect the FWHM of the resulting bilayer OECTs. This
parameter statistically decreases from ($P < 0.05$) 0.452, 0.216, 0.292,
0.188, to 0.113 V for BBL, NDI, BBL-PEDOT, NDI-PEDOT, and BBL-
P3CPT, respectively (Fig. 2d). Furthermore, P3CPT does not re-
dissolve in or be damaged by the acidic solvent for BBL, enabling the
top and bottom layers to be interchanged. Therefore, a P3CPT-BBL as
a bilayer vOECT was tested, showing a decreased response time but
no significant impact on anti-ambipolar characteristics (SI Note 1
and Fig. S8).

## Reconfigurable logic gate as artificial horizontal cells

In the human retina, the rods and cones photoreceptors are
responsible for light intensity and wavelength-dependent (color)
vision, respectively. The light level where both receptors are opera-
tional is called the mesopic range and includes most situations where
preprocessing of visual stimuli is essential, such as driving at night[38].
In this range, the signals produced by rods and cones are constantly
preprocessed by horizontal cells (HC) to encode specific informa-
tion, such as wavelength, intensity, and contrast. This preprocessing
occurs through lateral interactions in HCs, where a synaptic input
from cones provides a feedback output to both rods and cones[39].
Although signals in the human brain are not processed through
binary processes, the interdependent preprocessing of rods and
cones signals for wavelength and light intensity detection during
mesopic vision follows an association rule that can be replicated with
anti-ambipolar logic gates. Encoding via logic functions enables
many of the algorithms used in the retina to be implemented in
neuromorphic hardware, as the retinal preprocessing does not only
rely on spike encoded signal, but also on graded potentials
computing[14,30,38–42].

Here, reconfigurable logic circuits are developed exploiting the
high on/off ratio, and fast speed of BBL-PEDOT anti-ambipolar
vOECTs (Fig. 3a, d and SI Notes 2, 3). Two BBL-PEDOT devices are
combined in series to create either an AND logic gate or a NOR logic
gate, while an in-parallel combination of the same devices results in

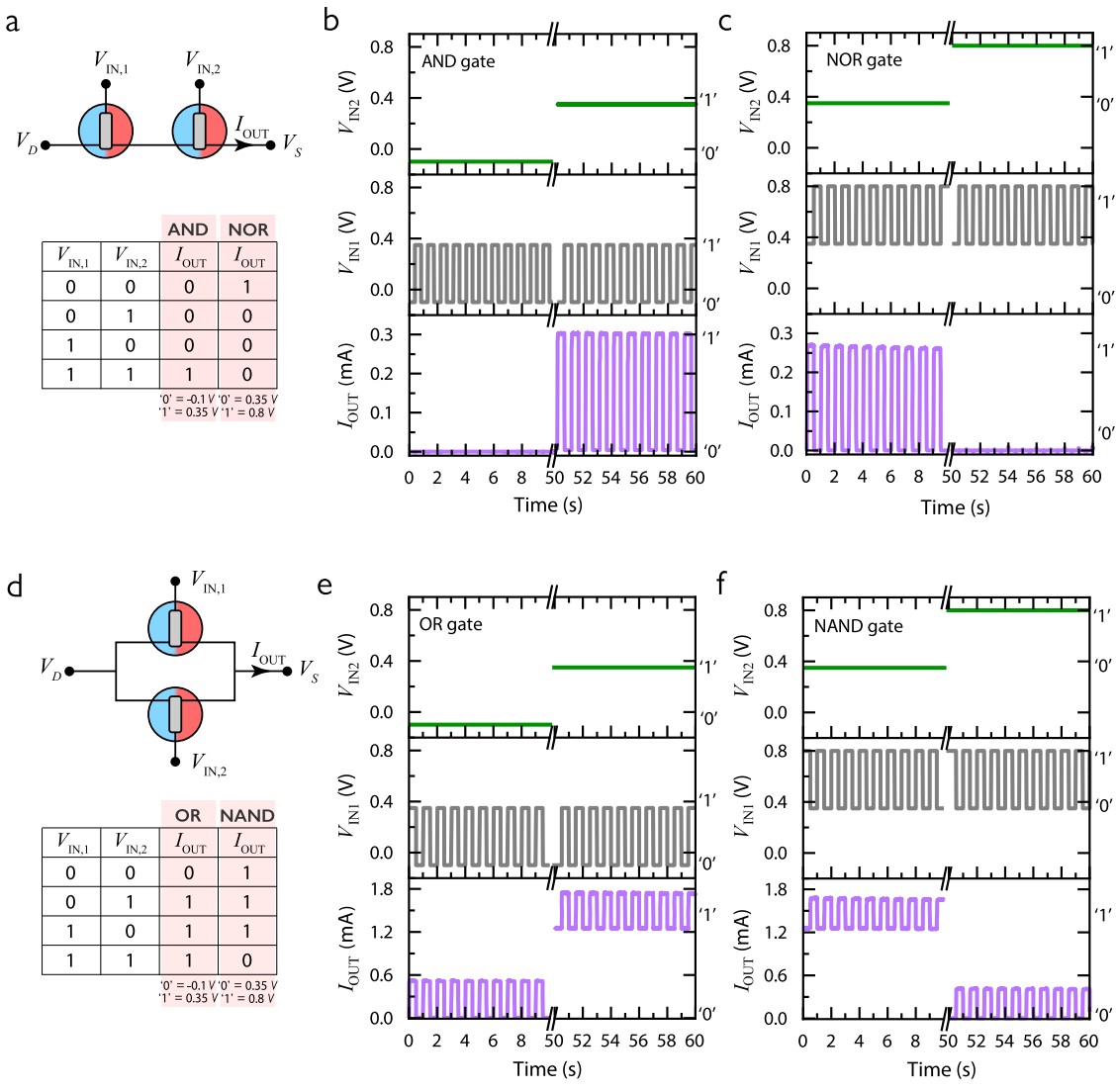

**Fig. 3 | Reconfigurable logic gates for bio-inspired signal preprocessing.**
**a** Circuit schematic for bilayer vOECT-based "AND/NOR" reconfigurable logic gate functions. A truth table for both states of the logic gate depending on the $V_{GS}$ definitions of "0" and "1". Both devices have a $W = 100\,\mu m$ and the applied $V_{DS}$ is 0.1 V. **b** Input voltages (green for $V_{IN2}$ and gray for $V_{IN1}$) and output current (purple) for the "AND" state of the logic gate. **c** Input voltages and output current for the

"NOR" state of the logic gate functions. **d** Circuit schematic for bilayer vOECT-based "OR/NAND" reconfigurable logic gate. A truth table for both states of the logic gate depending on the $V_{GS}$ definition of "0" and "1". Both devices have a $W = 100\,\mu m$ and the applied $V_{DS}$ is 0.1 V. **e** Input voltages and output current for the "OR" state of the logic gate. **f** Input voltages and output current for the "NAND" state of the logic gate.

either an OR logic gate or a NAND logic gate (Figs. 3a, d and S4). The logic gates associate two non-dependent inputs which are represented by the gate voltages $V_{IN1}$ and $V_{IN2}$, which can correspond to the light stimuli processed by the cones and rods, respectively. These stimuli are processed to deliver an output ($I_{OUT}$) corresponding to the $I_D$ of the logic gate circuit. Considering the AND logic gate, an output current is recorded only when both $V_{IN1}$ and $V_{IN2}$ are at 0.35 V which is assumed as the "true" logic state and corresponds to the peak position of the devices (Fig. 3b). However, by using 0.8 V (the second OFF state of the anti-ambipolar devices) as the "true" logic state, the negative transconductance is exploited leading to an NOR gate truth table (Fig. 3c). By using the same truth table assumptions but combining in-parallel the BBL-PEDOT devices, either an OR gate for the positive transconductance or a NAND gate for the negative transconductance are realized (Fig. 3e, f). As such, the reconfigurable logic gates enable a range of association rules replicating specific preprocessing functions, such as those done by horizontal cells, and are essential to be integrated into neuromorphic systems (SI Note 2 and Figs. S9–S13).

## Mimicking neuronal spiking via tailored Hodgkin–Huxley circuits

The Hodgkin–Huxley circuit for artificial neurons is the most bio-realistic neuron circuit as its components closely model the membrane ion channels characteristics generating action potentials, namely the potassium (K) and sodium (Na) channels (Fig. 4a). In biological neural systems, the Na channel acts by allowing ions passage when activated, thereby depolarizing the membrane potential ($V_{mem}$) and initiating the rising phase of an action potential. Specifically, the Na channel exhibits an inherent anti-ambipolarity and eventually "shuts off", while the repolarizing K channel current returns the membrane voltage to its resting state, thus creating an action potential[10,12,15] (Fig. 4a). In this context, the devices used for the Na channel control the spike threshold and frequency. Hence, matching these devices characteristics with the metrics of biological neurons is critical for the integration of these circuits in bio-hybrid systems.

Bilayer vOECTs are implemented for the first time in this model, to improve the performance of the circuit while specifically allowing for tuning of the spike threshold (SI Note 3). As such, the

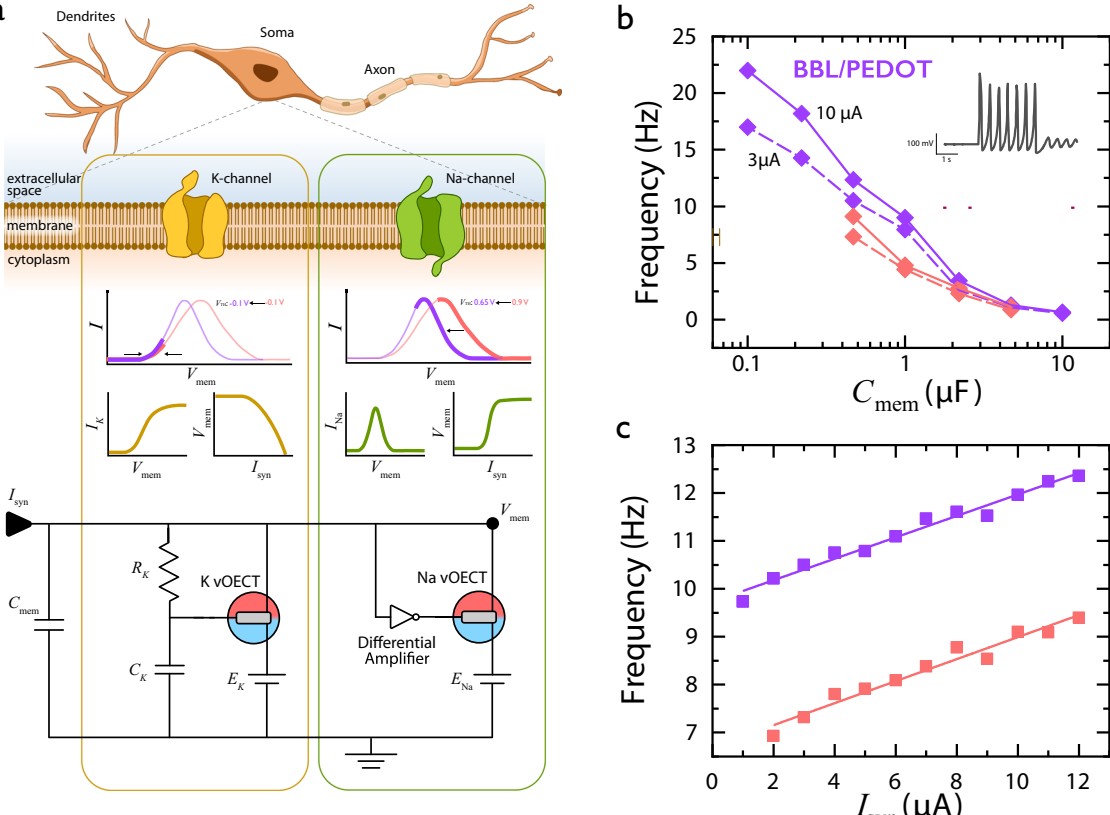

**Fig. 4 | Tailored Hodgkin–Huxley neurons. a** Diagram detailing the biological mechanisms behind action potential generation and the corresponding components in the HH circuit. For reference, representative transfer curves of BBL (red) and BBL-PEDOT (purple) are provided with the opaque region indicating the region active during circuit operation. Schematic adapted from ref. 10. **b** The change in frequency window with $C_{mem}$ compared between BBL ($W = 25\,\mu m$) and BBL-PEDOT ($W = 50\,\mu m$), where the solid trace represents the upper-frequency range, from

10 $\mu A$ of synaptic input current ($I_{syn}$), and the dotted trace represents the lower frequency range, from 3 $\mu A$ of $I_{syn}$. The inset shows representative spiking behavior during the current application. **c** The change in frequency with $I_{syn}$ at a $C_{mem}$ of 0.47 $\mu F$, where BBL/PEDOT (purple) exhibits a delta of 1.82 Hz and BBL (red) exhibits a delta of 1.785 Hz. To enable comparison between devices regardless of external circuit parameters such as $C_{mem}$, the lowest capacitance usable, i.e., the lowest value still eliciting a tonic spiking behavior, with the BBL vOECT is shown.

relevant metrics of HH circuits based on BBL-PEDOT bilayer vOECTs and on BBL vOECTs are presented and compared. In Fig. 4b, the circuit's spike frequencies show an exponential increase with decreasing values of the membrane capacitance ($C_{mem}$). However, the BBL-PEDOT HH circuit exhibits higher max frequency in the examined $C_{mem}$ range compared to the BBL-only HH circuit, despite the larger device size. Furthermore, the artificial neuron's spike frequency dependency on input current (external stimulus) is a measure to benchmark these circuits' biorealism and their potential integration into efficient computing systems. Hereby it is demonstrated that a bilayer HH circuit enables a larger spike frequency modulation, or range of frequencies under high and low inputs, regardless of the capacitance while enabling lower $C_{mem}$ values and thus higher frequencies, compared to single layer BBL circuits (Fig. 4b, c). The increased spike frequency is the result of the decreased response time and capacitance of the BBL-PEDOT bilayer vOECT compared to that of the BBL vOECT even at larger device sizes, enabling additional improvements in the future ($W = 25\,\mu m$ vs. $W = 50\,\mu m$) (SI Note 3). Indeed, this large frequency modulation window is key to perform spike-rate coding within neuromorphic systems to mimic the central nervous system computing functions. The tunable turn off of the bilayer vOECT approach also enables the modulation of the spiking threshold, while reducing device size and allowing close mimicry of biological neuron dynamics for future bio-hybrid coupling (Fig. 4a, SI Note 3, and Figs. S14–S18).

## Retinal pathway

In biology, anti-ambipolarity is prevalent in a number of biological systems beyond the neuronal Na channels, including rod photoreceptors' response to light intensity[32–34]. These photoreceptors show an anti-ambipolar behavior due to receptor bleaching of photopigments during dark adaptation, which results in an "OFF" response to high light intensity (Fig. 5a)[32–34]. The transduction of light proceeds through the photoreceptors to the horizontal cells, which are responsible for a range of preprocessing functionalities, including center/surround interactions, wavelength and intensity encoding, and global and local signal processing[29,30,32–34]. In this work the interaction between the photoreceptors and horizontal cell is replicated by coupling commercial light sensors with the BBL-PEDOT AND logic gate (Fig. 5b). Light from a collimated LED represents the external stimulus, activating two photodetectors working as rods and cones (Fig. 5a). Cone's wavelength specificity is replicated by using a green filter (550 ± 80 nm, Fig. S20) on the cone photodetector, while the rod one receives the unfiltered light input. Hence, the cone and the rod are connected to the $V_{IN1}$ and $V_{IN2}$ terminals of the logic gate AND, respectively (Fig. 5b). Operating as a horizontal cell, this logic gate combines the signals received from different photodetectors delivering a preprocessed voltage output. In cascade this output signal is coupled to the BBL-PEDOT spiking circuit, directly controlling its spiking activity (Fig. 5). Within this system, low light intensities (0–20 mW) correspond to the logic gate "False" state, leading to a silent HH neuron (Fig. 5c). Increasing light intensity leads to the

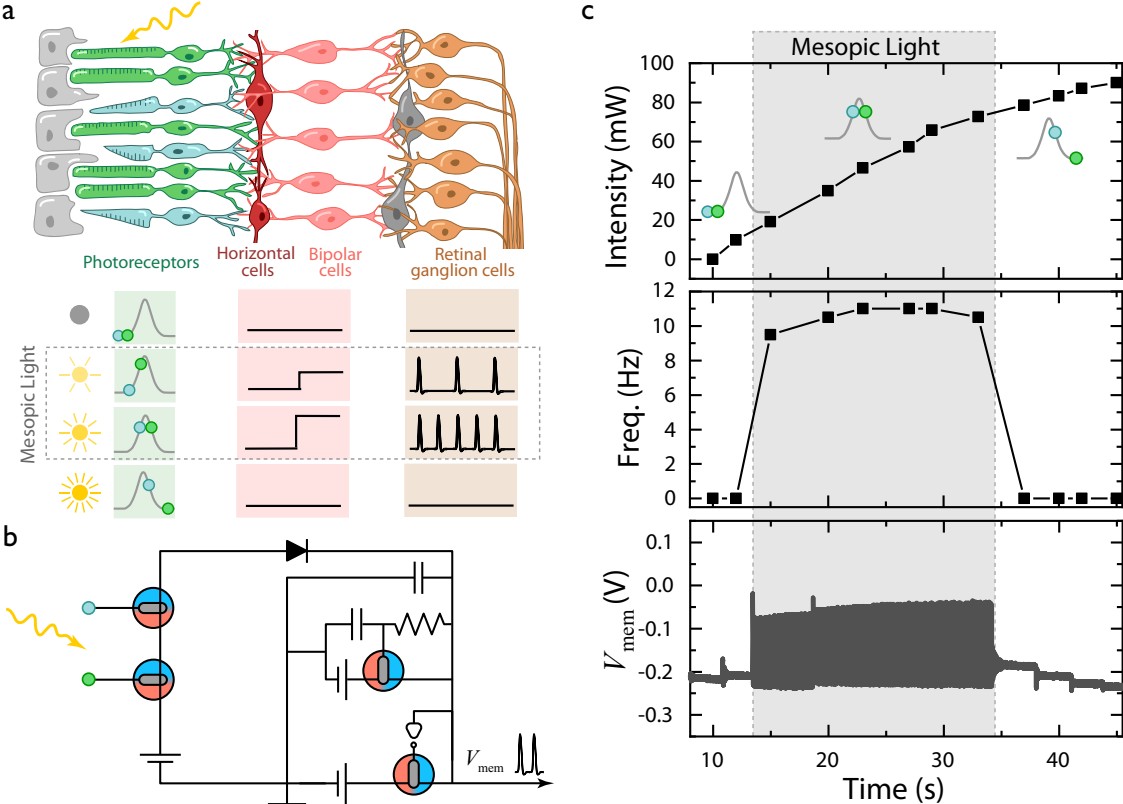

**Fig. 5 | Retinomorphic pathway using neuromorphic elements. a** Diagram illustrating the biological retinal pathway (top). Response of the logic gate devices to increasing light intensity and wavelength (bottom). **b** Equivalent circuit of the biological retinal pathway described in panel (**a**) built using $W = 50\,\mu m$ BBL/PEDOT devices. **c** Spiking output of the retinomorphic circuitry in response to the double stimuli represented by a dynamically increasing light intensity and the target wavelength (580 nm).

mesopic state, which fully activates the rod connected device, but only partially the cone device, resulting in low current output of the logic gate and so a low spike frequency (9 Hz) (Fig. 5c). When the light intensity reaches 40–70 mW, both the devices are in the ON-state, corresponding to a high current output of the artificial horizontal cell which triggers higher spike frequencies (11–11.5 Hz) (Fig. 5c). Finally, light intensities over 70 mW drive the $V_{INI}$ to the device's second "OFF" state leading to a zero current output, and replicating the biological bleaching of photoreceptors (Fig. 5c). Hence, a full retinal pathway from photoreceptor to horizontal cells to spiking neuron is replicated, emulating the preprocessing and encoding of wavelength and light intensity dependent signals (Fig. 5a). On this note, the inclusion of additional logic gates can add further preprocessing functions, serving to integrate and combine multiple stimuli converging on a single neuron/spiking circuit (SI Note 4 and Figs. S20–S32).

## Discussion

We have reported a general methodology for the fabrication of anti-ambipolar OECT transistors based on a vertical architecture where n-type and p-type materials are assembled as successive stacked bilayers. The bilayer approach enables control over the peak position, full width at half maximum, turn on, and turn off voltages of the vOECTs through material and thickness ratio selection. We demonstrated the versatility of the vOECT design proving its application as reconfigurable logic circuits and an HH-spiking circuit. Using such functions of bilayer devices, we replicated a retina-inspired pathway dependent on the anti-ambipolarity control of the vOECT. Our neuromorphic retinal pathway combines logic circuits which mimic the sensory preprocessing function of horizontal cells with HH neurons that act as retinal ganglion cells and encode the preprocessed signal

into precise spike patterns. The unique interplay of signal pre-processing and transmission elements, herald the design of increasingly sophisticated organic neuromorphic systems comprising adaptable and reconfigurable classification and filtering functions. As such, the vertical bilayer architecture formulates a novel concept for tunable organic electronic devices, with applications that are immediate in the organic neuromorphic field but encompass the fields of bioelectronics, wearable electronics, and informatics where device footprint, stability, design flexibility, and biocompatibility parameters are key.

Particularly, the material-dependent tunability of three-state logic circuits can be leveraged in the future to optimize these devices for minimal power consumption, switching speed, ion sensitivity, or bio-interfacing opening to a larger range of applications. The vertical architecture of the bilayer anti-ambipolar transistors indicates that the HH-spiking circuit can potentially be further reduced to minimal sizes. On the other hand, controlling vOECT characteristics allows for the fabrication of various neurons with differing threshold voltages, thereby mimicking the diversity of internal neuron characteristics present in the human brain and enabling a route toward the application of traditional neural microcircuits for a variety of computational tasks, such as sensory pathways and motor control[17–20]. For practical applications, the stability of the bilayers and integration density should be further developed and improved to enable the realization of reduced circuit sizes. In addition, synapses between photoreceptors, horizontal cells, and retinal ganglion cells are known to contribute significantly to early contrast enhancement, global and local signal processing, and key negative feedback mechanisms, and as such synaptic elements could be introduced to further develop this transduction system. This platform can serve as a building block

for the future integration of the various computational components of the retina, thereby enabling bio-interfaced, closed-loop, bidirectional retinal prosthetics. These platforms rely on the combination of spike encoding processed by spiking circuits (retinal ganglion cells) and graded potentials processed by logic circuits (horizontal cells), thus they not only overcome the biocompatibility and sensing limitations of current technologies but also promote a intrinsic neuromorphic computation paradigm through advanced anti-ambipolar devices[1,3,5,30,38–41].

## Methods

### Materials
The PEDOT:PSS solution (Clevios™ PH 1000, Heraeus) was mixed with 6.0 wt% ethylene glycol and 1.0 wt% (3-glycidyloxypropyl)trimethoxysilane all obtained from Sigma-Aldrich. BBL(Sigma-Aldrich) was dissolved in methanesulfonic acid (MSA) for a concentration of 5 mg ml$^{-1}$. The p(C6NDI-T)(SG303) was synthesized using the methods described in ref. 43, and dissolved in chloroform for a concentration of 20 mg ml$^{-1}$[43]. P3CPT(Reiki) was dissolved in dimethyl sulfoxide for a concentration of 15 mg ml$^{-1}$.

### Bilayer vertical OECT fabrication
Standard microscope glass slides (75 mm × 26 mm) were cleaned in a sonicator bath, first in DI water, then acetone, and finally in isopropanol. Gold bottom electrodes, with a thickness of 100 nm, were photolithographically patterned (with AZ-nLOF 2035 and SUSS-MJB4) on the cleaned glass slides. A titanium layer with a thickness of 12.5 nm was used to improve gold adhesion. A layer of Parylene C (SCS Coatings) was deposited with an adhesion promoter (Silane A-174 (γ-methacryloxypropyl trimethoxysilane), Sigma-Aldrich)) to reach a thickness of 1.36 μm. Soap (Micro-90 soap solution, 2% vol/vol in deionized water) was then spin-coated at 3500 rpm to separate the Parylene C layer from the following SU8-3010 layer. The SU8-3010 layer was spin-coated over the soap layer at 4000 rpm and then was photolithographically patterned (SUSS-MJB4). Following this, reactive ion etching (O$_2$/CHF$_3$ plasma, 160 W for 15 min with an O$_2$ flow rate of 50 s.c.c.m. and CHF$_3$ flow rate of 10 s.c.c.m.) was done to etch the Parylene C layer unprotected by SU8 thereby patterning an opening on the gold electrode for the channel material deposition. The BBL film was spin-coated in two steps, a 1 min 1000 rpm step immediately followed by a 15 s 3000 rpm step, then ethanol and DI water were spun at 3000 rpm to remove the MSA for a thickness of 85 nm. The PEDOT:PSS film was spin-coated at 1000 rpm for 1 min, dried at 100 °C for 5 min, and annealed at 135 °C for 45 min after peeling off SU8 for a thickness of 110 nm. The p(C6NDI-T)(SG303) film was spin-coated at 1000 rpm for 1 min to get a thickness of 200 nm. The P3CPT film was spin-coated at 1000 rpm for 1 min and dried via vacuum for 1 h to attain a thickness of 85 nm. The sacrificial SU8 layer was then peeled off to define the polymer within the device channels. For vertical OECTS, a third photolithographically patterning step (with AZ-nLOF 2035 and SUSS-MJB4) was done for the top gold electrode (100 nm). Finally, a passivating AZ-nLOF 2035 layer was defined using photolithography (SUSS-MJB4).

### Electrical characterization of OECTs
The current versus voltage characteristics of OECTs were gathered using a Keithley 2614B. Ag/AgCl pellets were used for gate electrodes with 100 mM PBS (phosphate-buffered saline) solution (Sigma-Aldrich) as an electrolyte. A second Keithley 2604B was used during Hodgkin–Huxley circuit characterization for the additional voltage sources.

### Retinal transduction pathway
A High-Power 3000 K Warm White LED Collimator Source and Compact Universal 2 channel LED Controller was used in conjunction with FD11A Si photodiodes purchased from MIGHTEX and Thor Labs, respectively. This combination was used to enable control over the light intensity applied to the pseudo-photoreceptors, which were composed of photodiodes and a supporting circuit. The cone photoreceptor was composed of a FD11A Si photodiode with 550 nm CWL, 80 nm FWHM optical filter (Edmund Optics) seated above it. Both photodiodes and the LED light source were seated in a 3D resin-printed chamber to minimize external light. The corresponding nodes were then measured using the aforementioned Keithleys and an Arduino with a measurement circuit, consisting of two operational amplifiers for converting the recorded signal into a positive voltage for the arduino to measure.

## Data availability
The original data underlying the figures in the main text are publicly available from the Northwestern University repository (Dryad) at https://doi.org/10.5061/dryad.44j0zpcpj.

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

## Acknowledgements

We would like to thank Dr. Ruiheng Wu for his help in understanding the XPS data of the bilayers, as well as Dr. Paul Smeets for his help regarding the FIB/SEM experiments. This work was supported by Northwestern's MRSEC, IRG-2 (NSF DMR-2308691). This work made use of the NUFAB facility of Northwestern University's NUANCE Center, which has received support from the SHyNE Resource (NSF ECCS-2025633), the IIN, and Northwestern's MRSEC program (NSF DMR-2308691). W.L. Leong gratefully acknowledges funding support from the Ministry of Education (MOE) under AcRF Tier 2 grant (MOE2019-T2-2-106) and the AcRF Tier 1 grant (RG118/21).

## Author contributions

Z.L., G.M.M., X.W., W.L.L. and J.R. conceptualized the research and established the theoretical approach. Z.L. and G.M.M. prepared the manuscript. Z.L. performed the experiments and analyzed the data regarding the BBL/PEDOT and NDI/PEDOT bilayers, Hodgkin–Huxley circuit implementation, and retinal pathway. Z.L. also developed simulations used in support of experimentation. A.S. and X.J. assisted in device design and fabrication optimization. X.W. and Z.Z. performed the experiments and analyzed the data for the P3CPT bilayers and logic circuits.

## Competing interests

The authors declare no competing interests.
