## [Peer Review File · Nature Communications]

Tunable Anti-Ambipolar Vertical Bilayer Organic Electrochemical Transistor enable Neuromorphic Retinal PathwayREVIEWER COMMENTS

Reviewer #1 (Remarks to the Author):

The paper presents an innovative OECT architecture exploiting a vertical layout to design a channel bilayer, composed of an n-type and a p-type organic semiconductors. Such designed v-OECT presents an anti-ambipolar transfer characteristic which can be tuned by tuning the bilayered-channel geometrical parameters and the materials employed. The authors propose to extract a general rule on how to predict the anti-ambipolar characteristic curve of a device given the bilayer parameters. With respect to Harikesh et al. 2023 work, the novelty here lies in the fact that the anti-ambipolarity tuning is done a priori based on the device design. The work is extensive and goes from the validation of the bilayer design concept, to the employment of such device in an H-H circuit and for retinal pathway biomimicking.

As a result, each paragraph of the main text is very condensed, limiting the clarity of the claims which sometimes seem poorly supported by the figures in the main text. In this sense the supplementary material is well made and fundamental to understand the work; this is especially true for the content of S1 Note 1.

Main text

Section 2.

Figure 1:

The numbering of the panels does not match the text.

Line 134. "The schematic in Figure 1a illustrates the structure of the bilayer vOECT" should be Figure 1c.

Line 139. Figure 1b should be recalled here when introducing the materials and showing their chemical structures.

Line 142. Figure 1a should be recalled here when discussing the variable resistors, instead of Figure 1b.

Line 144. Figure 1b recalled here is not informative according to the sentence.

Line 146. Provide a reference for XPS to confirm the results on PEDOT and BBL.

Line 154. Be explicit on what is considered a "lateral device". With reference to the Materials and Method section and to Figure S1, the architecture is not clear. Does the lateral device have a single layer channel patterned in between coplanar source and drain?

Line 166-168. This claim would be better supported by making explicit the explored W and L ratios, whose results are partially shown in figure S3.

Line 170. On which basis is the device defined stable? This statement does not agree with S1 NOTE 1 discussion and with the data presented in Figure S5.

Section 3.

Line 190. Clarify what is the "bilayer materials selection rule". This point is not clearly evident from section 2. A relevant discussion on this topic is S1 NOTE 1 that could be integrated in the main text.

Line 201. The transfer curves are normalized with respect to which parameter?

Section 4

Line 236. Typo: signals

Figure 3a and 3d. The figure shows the bilayer v-OECT 'AND/NOR' reconfigurable logic gate and their associated truth tables; there is no comparison with the biological counterpart.

Section 5.

Line 277. Avoid starting the sentence with "in biology".

Lines 277-281. Provide a reference.

Lines 282-284. The concept described in this sentence is not clear.

Line 285. Describe the implemented circuit.

Line 289. Explicit what Cmem stands for before using this abbreviation.

Line 291. Can the authors provide the rationale (even in the supplementary material) behind the choice of the device size? Why is the comparison held between devices of different sizes?

Line 293-295. Typo: H-H circuit enables; Rephrasing would make the sentence clearer. As it is now, the sentence applies to figure 4b, but in figure 4c for a the chosen value of Cmem BBL/PEDOT allows modulation at higher spiking frequencies, not a larger modulation in terms of

frequency range.

Figure 4a. The resolution is too low and the text too small to be properly readable. Moreover, Figure 4 is very similar to Figure 2 in "Harikesh, P.C., Yang, C.Y., Wu, H.Y. et al. Ion-tunable antiambipolarity in mixed ion-electron conducting polymers enables biorealistic organic electrochemical neurons. Nat. Mater. 22, 242–248 (2023)". If the purpose is to make a comparison with that work, explicit it in the text and cite it.

Figure 4b. Explicit what I_{syn} stands for before using this abbreviation. Also, in Figure S15, $I_{syn}=2\mu A$ is reported. Why in figure 4b $I_{syn}=3\mu A$ is reported for the lower current range?

Figure 4c. Make explicit why $C_{mem}=0.47\mu F$ value was chosen.

Section 6.

Recall Figure 5 and its sections in the text to guide the reading and understanding of this section.

Materials and methods.

Bilayer Vertical OECT Fabrication:

Line 390. What is the thickness of gold?

Line 392. What is the thickness of Parylene C?

Line 394. Clarify the role of SU8 in the process. A soap layer is introduced for its peeling and its patterning is described but it is not clear when and where is introduced in the process (also relevant with respect to clarity of Figure S1).

Line 396. In O_2/CHF_3 , numbers should be subscripts, here and later on.

Line 399. Explicit MSA.

Line 406-407. With reference to Figure S1, highlight AZ-nLOF 2035 passivating layer described here.

Be consistent with number and punctuation, in this section there are both 1,000 rpm and 3000 rpm (with and without comma). State the employed equipment for photolithography.

With reference to Figure S1, add a schematic to highlight the architectural difference between v-OECT and lateral OECT.

Supporting information

Note 1 Can the authors explicit which W values were investigated for clarity? (It can be derived from figure S3d).

Fig. S1. Please add a legend for the colors used in the schematics of the fabrication process, to identify the different materials in a more effective way. SU8 and parylene-C layers are not distinguished in the schematic. It's not clear what are the vertical walls.

Fig. S3b. Make explicit the parameters explored.

Fig. S3c. Overlay BBL transconductance curve to support the claim.

Fig. S8c. Two transfer curves per device type and condition indicated are reported, why?

Note 3

From the note: "Using this simulation, the impact of capacitance on spiking frequency was explored by reducing the capacitance of each component to a percentage of the original value as seen in Fig S17 c. By reducing the capacitances to 5% of the starting values ($C_{mem}=C_k=0.47\mu F$, $C_{oect} = 0.135\mu F$, $0.010\mu F$), the HH-spiking circuit is capable of producing spiking widths within the physiological range of 3-5ms, promising future central nervous system interfacing (Fig. S17)"

It is not clear the impact of which capacitance in the circuit was explored. Is it referred to C_{mem} or all of them? When conducting this exploration, were the starting values of the three starting capacitances reduced at the same time or were various combinations simulated?

For instance, it would help the discussion including the rationale behind the choice of the initial values and of the choice to reduce to 5%.

Fig. S14a. The circuit schematic is referred to as inverting amplifier, while in Figure 4a (main text) as differential amplifier. Coherence of terminology is preferable.

Fig. S16c. Add legend to make intuitive the distinction of the two traces. Add the time response of the BBL OECT as comparison to better support the claims in the note.

Figure S17c. Rephrase the sentence, not clear.

General comments

The number of samples N used to report statistical metrics (i.e. graphs reporting error bars or p-values) are often missing.

Specifically with reference to section 2 and section 3 and their relative supporting material, the various figures are based on devices of different dimensions (eg. Fig 1 $W=75\mu m$, Fig 2, Fig S5, Fig

S6 and Fig S8 W=100 um, Fig S4 W=50 um). Consistency would help comparison across experiments and figures. Such variability is particularly problematic for Figure S3 as in the same panel are shown output curve, transfer curve and transconductance for different device areas.

Reviewer #2 (Remarks to the Author):

This manuscript describes the design and fabrication of vertical, heterojunction OECTs made of p and n-type organic mixed conductors. The devices display non-linear electrochemical phenomena suitable for logic and neuromorphic applications. The advantage of this work is that the non-linear phenomena can be tuned precisely by changing the physical chemical properties of the interlayer. The work is interesting and important for the community. I believe that is suitable for Nature Communications. I only have a few comments in order to improve the quality of and readability:

- HH neurons are governed by very specific differential equations and usually require complex circuitry for implementation. Please soften the argument of HH neurons.
- The bilayer heterojunction films consist of a stack of p-type on top of n-type. How the reverse sequence of layers will impact device implementation and behavior in quasi-DC (IVs) and AC (spiking) characteristics?
- Is there a real heterojunction or is there any layer intermixing? Is it truly a linear combination of the materials characteristics?
- Retinal pathway. Band-pass firing is an important characteristic for showing bleaching and/or neural inhibition. However, in the actual visual system, input light intensity is mapped to firing frequency, which is not the case here. Please explain why this is not happening here and show or propose a way to induce it.
- A paragraph at the conclusions section can be added describing in which particular applications this system would be useful. What are the actual limitations of retinal implants?

Reviewer #3 (Remarks to the Author):

The authors present a general methodology for the fabrication of anti-ambipolar OECT by use of the p-type and n-type polymers. The vOECT shows tuning of anti-ambipolar characteristics that can perform logical functions and HH neurons. Using such functions of the devices, the authors replicated a retina-inspired pathway dependent on the anti-ambipolarity control of the vOECT. The following issues still need to be solved before publication to improve the quality of the paper.

1. Although vOECTs can emulate Na⁺/K⁺ ion channels, the HH neuron implemented by the vOECT-based circuit does not exhibit the more complex firing behaviors seen in HH neurons (refer to "Biological plausibility and stochasticity in scalable VO2 active memristor neurons," Nature Communications, 2018), and instead resembles the firing function of an LIF neuron. The authors should further improve the higher-order dynamic behavior of the neuron.
2. Compared with recent memristor/transistor-based HH neuron models (e.g., "Third-order nanocircuit elements for neuromorphic engineering," Nature, 2020; "Ion-tunable antiambipolarity in mixed ion-electron conducting polymers enables biorealistic organic electrochemical neurons," Nature Materials, 2023), what advantages do vOECT devices offer as neuromorphic devices in terms of device footprint and circuit complexity?
3. In Section 4, the authors highlight some retina-related functions but do not emphasize that "the mutual preprocessing of wavelength and light intensity detection signals from rod and cone cells follows an associative rule." They have constructed various vOECT-based circuits to realize logical functions, yet these do not fully capture the remarkable capabilities of visual processing. The importance of logical functions in visual information processing needs to be more strongly emphasized in the paper.
4. What about the stability of these vOECTs? This information would be crucial for bioinspired neuromorphic device applications.
5. The tunable threshold characteristic of the device is intriguing. The authors should discuss in detail the reasons for this tunable threshold and its impact on neuronal circuit control within the paper.
6. What is the significance of mimicking retinal pathways? The article should demonstrate specific applications (e.g., bio-interfaced closed-loop electronics).

Reviewer #4 (Remarks to the Author):

I co-reviewed this manuscript with one of the reviewers who provided the listed reports. This is part of the Nature Communications initiative to facilitate training in peer review and to provide appropriate recognition for Early Career Researchers who co-review manuscripts

In the manuscript and SI files, all changes are tracked using highlighting. Changes made in response to reviewer comments (text additions/changes and figure changes) are highlighted in yellow. Changes in response to journal/editorial style guidelines are highlighted in green.

REVIEWER COMMENTS

Reviewer #1 (Remarks to the Author):

The paper presents an innovative OECT architecture exploiting a vertical layout to design a channel bilayer, composed of an n-type and a p-type organic semiconductors. Such designed v-OECT presents an anti-ambipolar transfer characteristic which can be tuned by tuning the bilayered-channel geometrical parameters and the materials employed. The authors propose to extract a general rule on how to predict the anti-ambipolar characteristic curve of a device given the bilayer parameters. With respect to Harikesh et al. 2023 work, the novelty here lies in the fact that the anti-ambipolarity tuning is done a priori based on the device design.

The work is extensive and goes from the validation of the bilayer design concept, to the employment of such device in an H-H circuit and for retinal pathway biomimicking.

As a result, each paragraph of the main text is very condensed, limiting the clarity of the claims which sometimes seem poorly supported by the figures in the main text. In this sense the supplementary material is well made and fundamental to understand the work; this is especially true for the content of S1 Note 1.

Main text

Section 2.

Figure 1:

The numbering of the panels does not match the text.

We thank Reviewer 1 for taking the time and interest to read our manuscript in such detail, allowing us to correct these errors. We have revised the article as follows:

Line 134. "The schematic in Figure 1a illustrates the structure of the bilayer vOECT" should be Figure 1c.

We thank Reviewer 1, since our intention was to refer to Figure 1c. As such, we have adjusted the line to read: "The schematic in Figure 1c illustrates the structure of the bilayer vOECT." instead of "The schematic in Figure 1a illustrates the structure of the bilayer vOECT."

Line 139. Figure 1b should be recalled here when introducing the materials and showing their chemical structures.

We intended to refer to the chemical structure of the employed materials, as such have recalled Figures 1b.

Line 142. Figure 1a should be recalled here when discussing the variable resistors, instead of Figure 1b.

The intention was to refer to Figure 1a. As such line 142 has been updated to refer to Figure 1a instead of 1b.

Line 144. Figure 1b recalled here is not informative according to the sentence.

The intention was to refer to Figures 1c-d thereby highlighting the device design and bilayer structure as confirmed by XPS.

Line 146. Provide a reference for XPS to confirm the results on PEDOT and BBL.

We have provided three references #35-37(Wu, R., Paulsen, B. D., Ma, Q., McCulloch, I. & Rivnay, J. Quantitative Composition and Mesoscale Ion Distribution in p-Type Organic Mixed Ionic-Electronic Conductors. ACS Appl. Mater. Interfaces 15, 30553–30566 (2023), Kim, S.-M. et al. Influence of PEDOT:PSS crystallinity and composition on electrochemical transistor performance and long-term stability. Nat. Commun. 9, 3858 (2018), and Chen, Y. et al. In Situ Spectroscopic and Electrical Investigations of Ladder-type Conjugated Polymers Doped with Alkali Metals. Macromolecules 55, 7294–7302 (2022)) which serve to confirm the approach we followed to perform the elemental analysis of the XPS in accordance with the chemical structures shown in Figure 1b.

Line 154. Be explicit on what is considered a “lateral device”. With reference to the Materials and Method section and to Figure S1, the architecture is not clear. Does the lateral device have a single layer channel patterned in between coplanar source and drain?

We thank Reviewer 1 for the opportunity to clarify what is considered a “lateral device”. As per conventional OECT structures, lateral devices consist of “a single channel material connecting coplanar source and drain electrodes” as updated on Line 154.

Line 166-168. This claim would be better supported by making explicit the explored W and L ratios, whose results are partially shown in figure S3.

We have updated the text to better support our claim. To do so, we updated SI Note 1 to include the explicit values for W and L which have been considered:

“Across these tests, the W-values were varied from 25x25, 50x50, 75x75, to 100x100 μm , while the length of the PEDOT channel varied from 75 nm to 110 nm to 300 nm (Figure 1f-g, Figure S3).”

Line 170. On which basis is the device defined stable? This statement does not agree with S1 NOTE 1 discussion and with the data presented in Figure S5.

We tested the devices to 1200 cycles as illustrated in the updated Figure S5, which shows instability in FWHM, as caused by a decrease of the ON/OFF ratio that makes the device unstable past 750 cycles. We deemed a device unstable at a certain number of cycles, when due to a progressive degradation under operation, it could not be used anymore for the various applications shown in the paper. We have also added a short paragraph (included at the end of this response) to SI Note 1 to demonstrate this point and clarify that the decrease in peak position is not substantial enough to prevent usage of the device in the applications shown but could prevent other applications such as gaussian probabilistic networks from using this device as is. Furthermore, routes towards the improvement of stability is proposed but would primarily be dependent on the constraints of the application. This added text reads:

“Furthermore, as this layer is degraded under operation and also swells leading to increased contact resistance, the ON/OFF ratio of the device decreases until the FWHM becomes unstable, at which point the device is no longer usable for the applications demonstrated. As such, the stability of the bilayer vOECT is dependent on the application requirements, where the applications demonstrated do not require high stability in peak position but high stability in FWHM. Alternatively, a gaussian probabilistic network application would require minimal variation in peak position and FWHM over cycle count, and thus this system would need improvement in its stability to be a viable option.”

Section 3.

Line 190. Clarify what is the "bilayer materials selection rule". This point is not clearly evident from section 2. A relevant discussion on this topic is S1 NOTE 1 that could be integrated in the main text.

We are grateful to Reviewer 1 for the insightful comment which gives us the opportunity to clarify the "bilayer materials selection rule" as we feel it is a core component of the manuscript. This selection rule builds on the observation that an in-series summation of each vertical OECT layer's resistance can provide an approximation for the resulting bilayer transfer characteristic, thereby enabling a prediction of device performance before creation. As such, we have introduced a line within Section 2 to clearly present this predictive approach as the "bilayer materials selection rule", while discussing the in-series summation approximation of the BBL/PEDOT Bilayer device shown in Figure 1:

"Furthermore, summing the resistances of the BBL and PEDOT:PSS single layer vOECTs allows prediction of the resulting bilayer transfer characteristics, establishing a "bilayer materials selection rule" to guide the devices design and thus their performance across multiple material systems (Figure 1a,e, SI Note 1).

Line 201. The transfer curves are normalized with respect to which parameter?

Each transfer curve was normalized to its own peak I_{DS} , enabling fair shape comparison between devices of different current levels. We have amended the mentioned line, explaining that the transfer curves are self-normalized. This line now reads:

"Figure 2b illustrates the self-normalized transfer curves for each material used in the vOECT structure, with BBL and p(C6NDI-T) devices exhibiting anti-ambipolar characteristics."

Section 4

Line 236. Typo: signals

The word signal has been updated to say signals in line 236.

Figure 3a and 3d. The figure shows the bilayer v-OECT 'AND/NOR' reconfigurable logic gate and their associated truth tables; there is no comparison with the biological counterpart.

We thank Reviewer 1 for pointing out a potential source of confusion for this caption. We have adjusted the caption to simply read "Circuit schematic for", which eliminates the confusion.

Section 5.

Line 277. Avoid starting the sentence with "in biology".

Since we intend to draw the parallel with biological systems, we changed the incipit of the sentence into "In biological neural systems,".

Lines 277-281. Provide a reference.

We thank Reviewer 1 for their attention to detail and as such we have updated the manuscript to cite references #10,12,13 in Line 281. These references demonstrate other systems that utilize anti-ambipolar devices as an Na channel for spiking neurons, as well as Hodgkin and Huxley's original work demonstrating the OFF-ON-OFF nature of the Na channel in relation to action potential generation, while the K channel resets neurons after action potential generation.

Lines 282-284. The concept described in this sentence is not clear.

We acknowledge that the mentioned statement could be misinterpreted. We intended to explain that in biological neurons the sodium (Na) channel controls the spiking threshold due to its role in initiating the action potential. Accordingly, devices replicating Na channels in neuromorphic circuits also control the circuit's threshold and thus indirectly its frequency. Therefore, matching these devices characteristics to the metrics of biological Na Channels enables bio-inspired metrics (spike threshold and frequency), facilitating the integration of these circuit into bio-hybrid systems. As such, we have updated the line to read as follows for improved clarity:

"In this context, the devices used for the Na channel control the spike threshold and frequency. Hence, matching these devices characteristics with the metrics of biological neurons is critical for the integration of these circuits in bio-hybrid systems."

Line 285. Describe the implemented circuit.

We thank Reviewer 1 for their interest in the physical details of the spiking circuit experiment and have addressed this in the Supporting information within Note 3. There, we discussed the physical implementation of the circuit with breadboards and circuit elements. Additionally, on Line 285 of the main text a reference to SI note 3 is included to direct the reader towards that section if they have questions regarding the implementation of this system.

Line 289. Explicit what Cmem stands for before using this abbreviation.

We updated the manuscript on Line 289 to define Cmem as the membrane capacitance.

Line 291. Can the authors provide the rationale (even in the supplementary material) behind the choice of the device size? Why is the comparison held between devices of different sizes?

We thank Reviewer 1 for this comment which gives us the opportunity to avoid any potential cause of inaccuracy, reaffirming the role played by device size and the reasons that lead us to compare the characteristics of devices of different sizes.

Within the spiking circuit we utilized the 50x50 μm BBL/PEDOT devices due to the 25x25 μm BBL/PEDOT having an insufficient ON/OFF ratio to enable tonic spiking, while the 25x25 μm BBL device was used because it was the fastest BBL vOECT created. We found that 25x25 μm devices were the smallest device that could be reliably made in our academic cleanroom given the current fabrication protocols. As such this rationale has been added to SI Note 3 in the following text:

"Additionally, the 50x50 μm BBL/PEDOT Bilayer device was used instead of the 25x25 μm BBL/PEDOT device due to the insufficient ON/OFF ratio of the 25x25 μm device to enable reliable spiking. Furthermore, the 25x25 μm devices were the smallest device size created due to difficulties during peel-off in smaller devices, that lead to incomplete formation of the channel layers and thus electrical shorts between the top and bottom electrodes. Therefore, the 25x25 μm BBL devices were utilized in the vOECT spiking circuit due to their small footprint and increased speed compared to the other BBL vOECT dimensions (Figure S16).".

Additionally, we still hold the comparison between the devices of different sizes as it highlights the benefits of the bilayer approach (i.e. faster circuits and tunable threshold) at larger device sizes thereby providing increased motivation for employing bilayer devices. Indeed, a bilayer device that can be fabricated with a small W/L while retaining a sufficient ON/OFF ratio, would be increasingly superior for bio-integration applications (interfacing and power consumption arguments) as discussed in SI Note 3. This explanation is present in the main text within the following line, which we have updated to clarify this point:

"The increased spike frequency is the result of the decreased response time and capacitance of the

BBL-PEDOT bilayer vOECT compared to that of the BBL vOECT even at larger device sizes enabling additional improvements in the future ($W = 25 \mu\text{m}$ vs. $W = 50 \mu\text{m}$) (SI Note 3).”.

Line 293-295. Typo: H-H circuit enables; Rephrasing would make the sentence clearer. As it is now, the sentence applies to figure 4b, but in figure 4c for a the chosen value of C_{mem} BBL/PEDOT allows modulation at higher spiking frequencies, not a larger modulation in terms of frequency range.

We thank the reviewer for the opportunity to clarify this claim and improve the readability and understanding of the manuscript. As such, we have corrected the typo. With this statement, we intended to highlight that the bilayer based HH-circuits are compatible with lower C_{mem} , compared their BBL based counterpart, due to the bilayer’s decreased internal capacitance and response time., This allow these circuit to display a larger window of frequencies depending on the selected C_{mem} (ie 0.65 Hz to 22 Hz). Furthermore, the bilayer based HH-circuits also enable a slight increase in the window of modulation as seen in Figure 4c depending on I_{syn} . However, this difference is difficult to detect, and as such we have added the delta values into the caption. Additionally, the window is larger at higher C_{mem} values such as a C_{mem} of 1uF as seen in Figure 4b, further supporting our claims. Thus, we have rephrased the line to read as follows, to clarify any confusion:

“Hereby it is demonstrated that bilayer HH-circuit enables a larger spike frequency modulation, i.e. a larger dynamic range of frequencies is demonstrated depending on the current inputs, regardless of the capacitance, while also enabling the use of lower C_{mem} values and thus higher absolute frequencies, compared to single layer BBL circuits (Figure 4b,c).”.

Figure 4a. The resolution is too low and the text too small to be properly readable. Moreover, Figure 4 is very similar to Figure . 2 in “Harikesh, P.C., Yang, CY., Wu, HY. et al. Ion-tunable antiambipolarity in mixed ion–electron conducting polymers enables biorealistic organic electrochemical neurons. *Nat. Mater.* 22, 242–248 (2023)”. If the pourpose is to make a comparison with that work, explicit it in the text and cite it.

As per the resolution and text issues, a separate file was included providing the full-scale image without any distortion. While we understand there may are similarities between Figure 4 of the manuscript and the figure within Harikesh et al., the design of our figure was intended to be more broadly in line with the typical style of neuronal and Hodgkin-Huxley depictions seen in the literature, which have clearly become over the years a standard:

1 - Yi, W. et al. Biological plausibility and stochasticity in scalable VO₂ active memristor neurons. *Nat. Commun.* 9, 4661 (2018),

2 - Huang, H.-M. et al. Quasi-Hodgkin–Huxley Neurons with Leaky Integrate-and-Fire Functions Physically Realized with Memristive Devices. *Adv. Mater.* 31, 1803849 (2019),

3 - Hodgkin, A. L. & Huxley, A. F. A quantitative description of membrane current and its application to conduction and excitation in nerve. *J. Physiol.* 117, 500–544 (1952),

4 - Xu, Y., Gao, S., Li, Z., Yang, R. & Miao, X. Adaptive Hodgkin–Huxley Neuron for Retina-Inspired Perception. *Adv. Intell. Syst.* 4, 2200210 (2022).

This clear reference to a standardized schematic of the circuit design is intended to increase the immediacy of the discussion to the readers, allowing a clear understanding.

Additionally, our schematic is unique since our circuit employs bilayer and we have included (as element of novelty) the depictions of each device’s behavior during synaptic input.

That said, we appreciate the layout of elements in the Harikesh paper and are happy to acknowledge that our similar layout of figure elements is inspired by Harikesh (2023) in the figure caption.

Figure 4b. Explicit what I_{syn} stands for before using this abbreviation. Also, in Figure S15, $I_{syn}=2\mu A$ is reported. Why in figure 4b $I_{syn}=3\mu A$ is reported for the lower current range?

We thank the reviewer for their interest in the specifics of the spiking circuit application and its development. As such, we have specified I_{syn} within the caption as the input synaptic current, and the lowest value of this parameter (for which spikes can be triggered) depends on the value of the capacitance used in the circuit. Hence, for Figure S15, $I_{syn}=2\mu A$ for the lowest current input as the circuit at this capacitance is able to spike at lower current inputs due to the faster charging of the capacitor, but this is not the case for larger C_{mem} values, as such the data at $I_{syn}=3\mu A$ was used for Figure 4b due to its ability to generate spike at every capacitance used. To clarify this point, a line was included in SI Note 3 that reads:

"It is of interest to note that in Figure S15 the lowest I_{syn} generating spikes is $2\mu A$, whereas in Figure 4 the lowest is $3\mu A$. At the capacitance value used for this circuit, the faster charging of the capacitor enabled lower currents to generate spiking, which was not the case for all capacitors, and as such $3\mu A$ reliably produced spiking at every capacitance for the direct comparison seen in Figure 4. "

Figure 4c. Make explicit why $C_{mem}=0.47\mu F$ value was chosen.

We thank Reviewer 1 for their interest in the circuit parameters of the spiking circuit comparisons. We chose a C_{mem} value of $0.47\mu F$ for Figure 4c as it was the lowest capacitance at which the BBL circuit shows a tonic spiking behaviour. From this regime we demonstrated that higher frequencies can be achieved and we showed the circuit's frequency modulation window (for given C_{mem} values) by modulating I_{syn} . As such, the caption has been updated to read:

"The change in frequency with I_{syn} at a C_{mem} of $0.47\mu F$, where BBL/PEDOT (purple) exhibits a delta of 1.82 Hz and BBL (red) exhibits a delta of 1.785 Hz . To enable comparison between devices regardless of external circuit parameters such as C_{mem} , the lowest capacitance usable, i.e. the lowest value still eliciting a tonic spiking behavior, with the BBL vOECT is shown."

Section 6.

Recall Figure 5 and its sections in the text to guide the reading and understanding of this section.

We have updated Section 6 and included additional recollections to Figure 5 within multiple lines as highlighted within the text.

Materials and methods.

Bilayer Vertical OECT Fabrication:

Line 390. What is the thickness of gold?

For the gold thickness, we aimed to use the typically reported values, and as such both top and bottom electrodes were 100 nm thick.

Line 392. What is the thickness of Parylene C?

For the Parylene C layer, we use a thickness of $1.36\mu m$ to prevent shorting between top and bottom electrodes as well as enable easy peel off.

Line 394. Clarify the role of SU8 in the process. A soap layer is introduced for its peeling and its patterning is described but it is not clear when and where is introduced in the process (also relevant with respect to clarity of Figure S1).

We have updated the lines within the methods to read:

"Soap (Micro-90 soap solution, 2% vol/vol in deionized water) was then spin coated at 3500 rpm to separate the Parylene C layer from the following SU8-3010 layer. The SU8-3010 layer was spin coated over the soap layer at 4000 rpm and then was photolithographically patterned (SUSS-MJB4). Following this, reactive ion etching (O₂/CHF₃ plasma, 160 W for 15 min with O₂ flow rate of 50 s.c.c.m. and CHF₃ flow rate of 10 s.c.c.m.) was done to etch the Parylene C layer unprotected by SU8 thereby patterning an opening on the gold electrode for the channel material deposition."

These updated lines clarify that the SU8 layer is cast on top of the soap layer, and enables patterning of vOECT channels by defining the opening of the bottom electrode via photolithography and etching. Additionally, the SU8 layer acts as a sacrificial layer to remove material outside the channel layer during the peel-off step similar to conventional co-planar OECT fabrication procedures. In addition, we have updated Figure S1 based on this and other points to clearly identify each step of the fabrication process and the materials used.

Line 396. In O₂/CHF₃, numbers should be subscripts, here and later on.

We have appropriately updated the numbering to act as subscripts in all instances.

Line 399. Explicit MSA.

We thank Reviewer 1 for exploring the methods with such detail. In this instance, MSA is referring to methanesulfonic acid and we have updated the article to include this in line 384.

Line 406-407. With reference to Figure S1, highlight AZ-nLOF 2035 passivating layer described here. Be consistent with number and punctuation, in this section there are both 1,000 rpm and 3000 rpm (with and without comma). State the employed equipment for photolithography. With reference to Figure S1, add a schematic to highlight the architectural difference between v-OECT and lateral OECT.

We have incorporated the reviewers suggestions to improve the quality of the manuscript. For the consistency of punctuation, all version have been changed to the format without the comma and the equipment used for the photolithography. For the updates to Figure S1, these points have been incorporated and addressed as described below, but in summary a comparison has been included between vertical and lateral OECT structures, and the passivation layer has been included in the legend.

Supporting information

Note 1 Can the authors explicit which W values were investigated for clarity? (It can be derived from figure S3d).

We thank the reviewer for their interest in the device sizes explored and as such have included a line within SI Note 1 to explicitly state which W values were investigated along with L values were investigated in figure S3. This line reads as follows:

"Across these tests, the W-values were varied from 25x25, 50x50, 75x75, to 100x100 μm, while the length of the PEDOT channel varied from 75 nm to 110 nm to 300 nm (Figure 1f-g, Figure S3)."

Fig. S1. Please add a legend for the colors used in the schematics of the fabrication process, to identify the different materials in a more effective way. SU8 and parylene-C layers are not distinguished in the schematic. It's not clear what are the vertical walls.

We thank the reviewer for their insightful suggestion and as such have updated Figure S1, based on this and the aforementioned comments. As such, we have instituted a unique color for each material and have expanded the caption to include additional details as well as a label for each color used in the schematic. Additionally, we have added panels g and h to provide a schematic of the architectural differences between vOECTs and conventional lateral OECTs.

Fig. S3b. Make explicit the parameters explored.

We have updated the caption for Figure S3b to read:

"Example devices for changing the channel length ratio (ie the length of PEDOT layer/length of BBL layer) of the BBL/PEDOT bilayer vOECT as discussed in Section 2 and Figure 1 ($W=75\ \mu\text{m}$, $V_{DS}=0.15\text{V}$)", thereby explicitly stating the parameters explored in as clear a manner as possible.

Fig. S3c. Overlay BBL transconductance curve to support the claim.

We thank Reviewer 1 for the opportunity to improve this manuscript by improving the support for the claims made. As such, we have incorporated your suggestion and overlaid a BBL vOECT transconductance to Figure S3, while updating the caption.

Fig. S8c. Two transfer curves per device type and condition indicated are reported, why?

We thank the reviewer for their question and have clarified the caption for Figure S8c. In this figure two transfer curves per device type are not being shown, rather a single transfer curve with both increasing V_{GS} and decreasing V_{GS} is shown as that measurement is the standard in the OECT field for displaying transfer curves, while showing the hysteresis. As such, we have updated the caption to read as such:

"Corresponding transfer curves of bilayer devices with the layers swapped in both increasing and decreasing VGS conditions to demonstrate hysteresis. In both devices the increasing VGS curve reached a higher peak current than the decreasing VGS curve ($W=100\ \mu\text{m}$, $V_{DS}=0.3\text{V}$)."

Note 3

From the note: "Using this simulation, the impact of capacitance on spiking frequency was explored by reducing the capacitance of each component to a percentage of the original value as seen in Fig S17 c. By reducing the capacitances to 5% of the starting values ($C_{mem}=C_k=0.47\ \mu\text{F}$, $C_{oect} = 0.135\ \mu\text{F}$, $0.010\ \mu\text{F}$), the HH-spiking circuit is capable of producing spikes widths within the physiological range of 3-5ms, promising future central nervous system interfacing (Fig. S17)"

It is not clear the impact of which capacitance in the circuit was explored. Is it referred to C_{mem} or all of them? When conducting this exploration, were the starting values of the three starting capacitances reduced at the same time or were various combinations simulated?

For instance, it would help the discussion including the rationale behind the choice of the initial values and of the choice to reduce to 5%.

We thank Reviewer 1 for pointing out this potential source of confusion thus enabling us to clarify this statement and its intention. We reduced simultaneously the value of all the capacitances (C_{mem} , C_{oect} , C_k) in the simulation following a half order of magnitude steps (from 100% to 50%, 50% to 10%, 10% to 5%). As evident from these simulations, a physiological range of spike widths was achieved when capacitances values corresponding to 5% of the initial starting values for all the capacitors was employed (simultaneous change). To avoid any confusion, we updated SI Note 3:

"Using this simulation, the impact of capacitance on spiking frequency was explored by reducing the capacitance of every component simultaneously to a percentage of the original value as seen in Figure

S17 c. By reducing the capacitances in half order of magnitude steps (ie 50, 10, 5, 1) of the starting values ($C_{mem}=C_k=0.47\mu F$, $C_{oect} = 0.135\mu F$, $0.010\mu F$), the HH-spiking circuit replicates spikes widths within the physiological range of 3-5ms at 5% of the initial capacitance values, promising future central nervous system interfacing (Figure S17)18,20,23."

Fig. S14a. The circuit schematic is referred to as inverting amplifier, while in Figure 4a (main text) as differential amplifier. Coherence of terminology is preferable.

We thank Reviewer 1 for identifying this incoherence, and as such have updated the caption for Figure S14a to read differential operational amplifier.

Fig. S16c. Add legend to make intuitive the distinction of the two traces. Add the time response of the BBL vOECT as comparison to better support the claims in the note.

We have included the response time from an example BBL vOECT in Figure S16c. We have updated the caption to provide a distinction between the traces and as such the caption now reads:

" c, Example response time measurement of a BBL/PEDOT bilayer vOECT (IDS is light purple, VGS is dark purple) and BBL vOECT (IDS is light red, VGS is dark red) ($W=50\ \mu m$, $V_{DS} = 0.15\ V$). "

Figure S17c. Rephrase the sentence, not clear.

We have updated the caption to read:

"The high and low input frequencies are displayed as all capacitances within the circuit are simultaneously reduced to the corresponding percentage of their initial values."

General comments

The number of samples N used to report statistical metrics (i.e. graphs reporting error bars or p-values) are often missing.

Specifically with reference to section 2 and section 3 and their relative supporting material, the various figures are based on devices of different dimensions (eg. Fig 1 $W=75\ \mu m$, Fig 2, Fig S5, Fig S6 and Fig S8 $W=100\ \mu m$, Fig S4 $W=50\ \mu m$). Consistency would help comparison across experiments and figures. Such variability is particularly problematic for Figure S3 as in the same panel are shown output curve, transfer curve and transconductance for different device areas.

We thank the Reviewer for their careful consideration of our manuscript and the insightful and constructive suggestions. We have updated all the figures to include the sample size and p-values where relevant (in case of missing error bars). Additionally, we have updated the figures containing error bars to display larger error bars thus making it more visible and more evident for the readers.

To keep consistency among the dimensions of the different devices compared in the manuscript, we have updated the data presented in Figure 1 which are now extracted from a $W=100\ \mu m$ device. In Figure S4, we also provide an exemplary fitting for a $W=50\ \mu m$ BBL/PEDOT device since this device served as our internal reference for the analysis of the spiking circuit and its simulations, and thus the capacitance of this device has been thoroughly examined. As for Figure S3, we have updated panel A and C to represent the data of a $W=100\ \mu m$ device. However, for panel B, not reproducible data have been collected to properly present $W=100\ \mu m$ devices, due to inhomogeneous formation of a thick PEDOT layer on larger devices.

Reviewer #2 (Remarks to the Author):

This manuscript describes the design and fabrication of vertical, heterojunction OECTs made of p and n-type organic mixed conductors. The devices display non-linear electrochemical phenomena suitable for logic and neuromorphic applications. The advantage of this work is that the non-linear phenomena can be tuned precisely by changing the physical chemical properties of the interlayer. The work is interesting and important for the community. I believe that is suitable for Nature Communications. I only have a few comments in order to improve the quality of and readability:

- HH neurons are governed by very specific differential equations and usually require complex circuitry for implementation. Please soften the argument of HH neurons.

A significant advantage of OECT-based HH circuits is that they enable reduced circuitry complexity for in-hardware implementation. As demonstrated in "Ion-tunable antiambipolarity in mixed ion–electron conducting polymers enables bio-realistic organic electrochemical neurons," Nature Materials, 2023), OECT-based HH-circuits preserve the complex spiking behaviors of conventional electronics HH- implementations, including factors such as rebound spiking, accommodation, spike latency, input integration, class 1 spiking, class 2 spiking, phasic spiking, phasic bursting, refractory periods, variable thresholds, and resonance. In our manuscript, to demonstrate that bilayer vOECT based HH-circuits retain these capabilities we have included simulations, using the same model as within other sections of the SI, showing these properties within Figure S19. Additionally, we have included text within SI Note 3 to highlight that these properties remain despite our main text focusing primarily on firing rate encoding. This text reads as follows:

"Finally, the range of known OECT-based HH neuron characteristics is simulated using the circuit in Figure S17, thereby confirming the perseverance of complex firing behaviors in the bilayer vOECT neurons despite primarily using tonic firing (Figure S19). Further advancements in the complexity of spiking behaviors in OECT-based HH neurons could be achieved via inclusion of additional input circuitry as demonstrated in Yi et al. as well as improvements in p-type material selection, which can be optimized to increase threshold stochasticity and include adaptive threshold capabilities, such as those demonstrated in PEDOT:PTHF15,28."

As for the relationship between the devices used and the set of differential equations governing HH neurons, this has been previously discussed in Harikesh et al., and we here reiterate the discussion to prove its validity in the context of our system. The intrinsic properties of the anti-ambipolar OECT enable spiking via replication of the key activation and inactivation parameters of the Na and K channels presented in the classical Hodgkin-Huxley equations. These equations, Equation 1 and Equation 2, are as follows:

$$I_{syn}(t) = C_{mem} \frac{dV_{mem}}{dt} - \sum_j I_j(t) \quad (1)$$

$$\sum_j I_j(t) = g_K n^4 * (V_{mem} - E_K) + g_{Na} m^3 h (V_{mem} - E_{Na}) + g_L (V_{mem} - E_L) \quad (2)$$

where C_m represents the membrane capacitance in both systems, E_K , E_{Na} , and E_L are the channel reversal potentials equivalent to voltage sources, and m , n , and h are ion channel gating variables. These gating variables relate the voltage of the membrane to the conductance of each channel, where m represents the activation of the Na channel, h represents the inactivation of the Na channel, and n represents the activation of the K channel. Thus, the bilayer anti-ambipolar OECTs can replicate the relationship between these gating variables and the conductance of the channel due to the OFF-ON-

OFF behavior of the device's transfer curve, and the OFF-ON behavior of the Gate-Drain connected voltage sweep (Figure 4). Specifically, the K channel behavior can be replicated with the traditional linear regime OECT equations, as seen in equation 4, where the activation gating variable is akin to the current dependence on applied and threshold voltage. Additionally, the Na channel behavior can be replicated with two equations, Equation 5 and Equation 6, where each equation replicates the activation and inactivation gating variable based on the changes in mobility and threshold voltage when switching from OFF to ON and ON to OFF, respectively. These equations, Equation 3, Equation 4, Equation 5, and Equation 6, previously presented in Harikesh et al. are as follows:

$$I_{syn}(t) = C_{mem} \frac{dV_{mem}}{dt} + i_K(t) - i_{Na}(t) \quad (3)$$

$$i_K(t) = \mu_K C \frac{W*d}{L} * \left((V_{mem}(1 - e^{\frac{t}{RC}}) - E_K - V_K) (V_{mem} - E_K) - \frac{(V_{mem} - E_K)^2}{2} \right) \quad (4)$$

$$i_{Na}(t) = \mu_{Nam} C \frac{W*d}{L} * (f(V_{mem}, t) - V_{mem} - V_{Nam})^2, \left(V_{Nam} < f(V_{mem}) < \frac{(V_{Nam} - V_{Nah})}{2} \right) \quad (5)$$

$$i_{Na}(t) = \mu_{Nah} C \frac{W*d}{L} * (f(V_{mem}, t) - E_{Na} - V_{Nah})^2, \left(\frac{(V_{Nam} - V_{Nah})}{2} < f(V_{mem}) < V_{Nah} \right) \quad (6)$$

where $f(V_{mem}, t)$ is the output of the amplifier circuit^{2,20,21}. These equations relate the inherent capacitances (C), mobilities (μ), thresholds (V), and device dimensions (W, d, L) of each channel's device to its corresponding output current, enabling OFF-ON behavior in the K channel and OFF-ON-OFF behavior in the Na channel^{2,20,21}. Thus, the behaviors defined by the Hodgkin-Huxley equations are replicated. Lastly, the leakage channel is simply replicated by the device leakage.

However, since this is not novel, we have included a simplified explanation and a referral to the work done by Harikesh et al. for an in-depth explanation. These lines are as follows:

"As previously discussed in Harikesh et al., the intrinsic properties of the anti-ambipolar OECT enable replication of neuron spiking activity by mimicking the key activation and inactivation parameters of the Na and K channels presented in the classical Hodgkin-Huxley equations^{2,20,21}. The bilayer anti-ambipolar OECTs can replicate the relationship between these gating variables and the conductance of the channel due to the OFF-ON-OFF behavior of the device's transfer curve, and the OFF-ON behavior of the Gate-Drain connected voltage sweep (Figure 4). Specifically, the K channel behavior can be replicated with the traditional linear regime OECT equations where the activation gating variable is akin to the current dependence on applied and threshold voltage^{2,20,21}. On the other hand, the Na channel behavior can be replicated as an accumulation mode and depletion mode material if VGS is before or after the peak, respectively. This then replicates the activation and inactivation gating variable based on the changes in mobility and threshold voltage when switching from OFF to ON and ON to OFF, respectively^{2,20,21}. For a detailed explanation of the relationship between device equations and the Hodgkin-Huxley equations, we refer the reader to Supplementary Notes 3 and 4 within Harikesh et al^{2,20,21}."

- The bilayer heterojunction films consist of a stack of p-type on top of n-type. How the reverse sequence of layers will impact device implementation and behavior in quasi-DC (IVs) and AC (spiking) characteristics?

We thank reviewer 2 for their interest in the bilayer structure and the influence of the stack order on the device's functionality. As mentioned in Section 2, Section 3, SI Note 1, and Figure S8, most p-type materials are not compatible with the solvents used to cast n-type materials. In our manuscript's case the n-type solvent for BBL is methanesulfonic acid which potentially leads to acid crystallization or degradation of p-type materials films. Thus, in case of BBL/PEDOT bilayers the only viable fabrication procedure was to deposit first the n-type material. However, in case of BBL/P3CPT bilayers we were able to fabricate devices with reverse materials casting order. As such, the devices consisting of the n-

type on top of the p-type are displayed in Figure S8 and exhibit similar properties as the reverse order, with the primary exception of device speed. Since the n-type (BBL) is more hydrophobic than the p-type (P3CPT), the response time increased leading to minor shifts in threshold voltage and peak position at the same scan speed due to the n-type layer exhibiting ion transport first while hindering ion transport into the p-type layer, as mentioned in SI Note 1. To describe how this altered structure would affect spiking characteristics we added a line to the end of SI Note 1:

“For example, the increased response time of this reversed stack would lead to slower switching between logic states in the case of anti-ambipolar logic circuits, or lower spiking frequencies in the case of anti-ambipolar driven spiking circuits.”

- Is there a real heterojunction or is there any layer intermixing? Is a truly a linear combination of the materials characteristics?

As mentioned in SI note 1, the multiple spin coating steps leading to the formation of the bilayer determine a small intermixing region (10-20nm) at the interface of the BBL and PEDOT stacked films, as shown by XPS. Furthermore, Figure 1e demonstrates a linear combination of the individual layer resistances, which matches closely with recorded devices characteristics, as confirmed also by the EIS characteristics demonstrated in S4. To avoid any confusion relating to the nature of this intermixing and on the function of the bilayer, we included a line in SI Note 1:

“This narrow intermix area may improve ion intercalation, promoting the penetration of ions through PEDOT:PSS into BBL films, but as demonstrated in Figure 1e, Figure S3, and Figure S4 this phenomenon has minimal impact on device characteristics in comparison to the bulk properties of each layer.”

That said, these findings are likely highly materials dependent, and may not be generalizable to all materials.

- Retinal pathway. Band-pass firing is important characteristic for showing bleaching and/or neural inhibition. However, in the actual visual system, input light intensity is be mapped in firing frequency, which is not the case here. Please explain why this in not happening here and show or propose a way to induce it.

We thank the reviewer for their interest in expanding the bilayer vOECT applications within the visual system. While we primarily show bandpass firing, there is a slight modulation in firing frequency during the ON-state indicating that input light intensity can be mapped into firing frequency. Indeed, we show in SI Note 4 and Figure S25-30 that light intensity can modulate the firing frequency. However, since the BBL/PEDOT Bilayer is symmetrical, the band-pass firing capabilities overshadow the mapping capabilities. Therefore, if a bilayer were designed to be asymmetrical, with a flat linear transfer curve leading to a sharp cut off, similar to a triangle waveform, then the increasing light intensity would map directly to firing frequency much more accurately, while the bandpass capabilities would continue to exist. As such, this would have required additional material testings, which we feel were out of the scope of this manuscript. To implement the advice included in the reviewer’s comment, we have included a description of the above points within SI Note 4.

- A paragraph at the conclusions section can be added describing in which particular applications this system would be useful. What are the actual limitation of retinal implants?

We thank Reviewer 2 for the constructive criticism of our manuscript and the opportunity to discuss this in more detail. As discussed in the Conclusions Section, the bilayer vOECT system obvious applications target reconfigurable logic circuits and the simplification of spiking neurons, by increasing single device computational complexity to reduce spatial cost and device count per circuit. Furthermore, we discuss directions that can lead towards larger scale systems such as retinal

prosthetics as well as discuss limitations and improvements necessary to reach such systems. To further discuss the applications of the retinal pathway as well as highlight the current biosensing and biocompatibility limitations of current retinal prosthetics, we included the following within the conclusion:

"This system can serve as a building block for the future integration of the various computational components of the retina thereby enabling bio-interfaced, closed loop, bidirectional retinal prosthetics. These systems rely on the combination of spike encoding processed by spiking circuits (retinal ganglion cells) and graded potentials processed by logic circuits (horizontal cells), thus they not only overcome the biocompatibility and sensing limitations of current technologies but also promote an intrinsic neuromorphic computation paradigm through advanced anti-ambipolar devices^{1,3,5,30,38-41}."

With this line, we included two additional references (Wu, K. Y., Mina, M., Sahyoun, J.-Y., Kalevar, A. & Tran, S. D. Retinal Prostheses: Engineering and Clinical Perspectives for Vision Restoration. *Sensors* 23, (2023)., Matrone, G. M. et al. A modular organic neuromorphic spiking circuit for retina-inspired sensory coding and neurotransmitter-mediated neural pathways. *Nat. Commun.* 15, 2868 (2024)) that review retinal prosthetics and demonstrate additional OECT retinal pathways, respectively, to support the claims made.

Reviewer #3 (Remarks to the Author):

The authors present a general methodology for the fabrication of anti-ambipolar OECT by use of the p-type and n-type polymers. The vOECT shows tuning anti-ambipolar characteristics that can perform logical function and HH neuron. Using such functions of the devices, the authors replicated a retina-inspired pathway dependent on the anti-ambipolarity control of the vOECT. The following issues still need to be solved before publication to improve the quality of the paper.

1. Although vOECTs can emulate Na⁺/K⁺ ion channels, the HH neuron implemented by the vOECT-based circuit does not exhibit the more complex firing behaviors seen in HH neurons (refer to "Biological plausibility and stochasticity in scalable VO₂ active memristor neurons," *Nature Communications*, 2018), and instead resembles the firing function of an LIF neuron. The authors should further improve the higher-order dynamic behavior of the neuron.

We thank the reviewer for their interest in the complex firing behaviors of our vOECT HH neurons. As previously shown in "Ion-tunable antiambipolarity in mixed ion–electron conducting polymers enables biorealistic organic electrochemical neurons," *Nature Materials*, 2023", anti-ambipolar OECT based HH circuits are capable of replicating a wide range of complex HH neuron behaviors without the additional circuitry presented within "Biological plausibility and stochasticity in scalable VO₂ active memristor neurons," *Nature Communications*, 2018), which could certainly be implemented as an input modification layer before the spiking circuit to attain even more complex functionalities. As such, we have included a section within SI Note 3 to discuss this opportunity and the opportunity for additional material optimization, as well as including simulations within Figure S19 to demonstrate a range of HH behaviors retained in the bilayer vOECTs from BBL-based lateral anti-ambipolar OECT HH circuits.

As for the relationship between the devices used and the set of differential equations governing HH neurons, this has been previously discussed in Harikesh et al., and we here reiterate the discussion to prove its validity in the context of our system. The intrinsic properties of the anti-ambipolar OECT enable spiking via replication of the key activation and inactivation parameters of the Na and K channels presented in the classical Hodgkin-Huxley equations. These equations, Equation 1 and Equation 2, are as follows:

$$I_{syn}(t) = C_{mem} \frac{dV_{mem}}{dt} - \sum_j I_j(t) \quad (1)$$

$$\sum_j I_j(t) = g_K n^4 * (V_{mem} - E_K) + g_{Na} m^3 h (V_{mem} - E_{Na}) + g_L (V_{mem} - E_L) \quad (2)$$

where C_m represents the membrane capacitance in both systems, E_K , E_{Na} , and E_L are the channel reversal potentials equivalent to voltage sources, and m , n , and h are ion channel gating variables. These gating variables relate the voltage of the membrane to the conductance of each channel, where m represents the activation of the Na channel, h represents the inactivation of the Na channel, and n represents the activation of the K channel. Thus, the bilayer anti-ambipolar OECTs can replicate the relationship between these gating variables and the conductance of the channel due to the OFF-ON-OFF behavior of the device's transfer curve, and the OFF-ON behavior of the Gate-Drain connected voltage sweep (Figure 4). Specifically, the K channel behavior can be replicated with the traditional linear regime OECT equations, as seen in equation 4, where the activation gating variable is akin to the current dependance on applied and threshold voltage. Additionally, the Na channel behavior can be replicated with two equations, Equation 5 and Equation 6, where each equation replicates the activation and inactivation gating variable based on the changes in mobility and threshold voltage when switching from OFF to ON and ON to OFF, respectively. These equations, Equation 3, Equation 4, Equation 5, and Equation 6, previously presented in Harikesh et al. are as follows:

$$I_{syn}(t) = C_{mem} \frac{dV_{mem}}{dt} + i_K(t) - i_{Na}(t) \quad (3)$$

$$i_K(t) = \mu_K C \frac{W*d}{L} * \left((V_{mem}(1 - e^{\frac{t}{RC}}) - E_K - V_K) (V_{mem} - E_K) - \frac{(V_{mem} - E_K)^2}{2} \right) \quad (4)$$

$$i_{Na}(t) = \mu_{Nam} C \frac{W*d}{L} * (f(V_{mem}, t) - V_{mem} - V_{Nam})^2, \left(V_{Nam} < f(V_{mem}) < \frac{(V_{Nam} - V_{Nah})}{2} \right) \quad (5)$$

$$i_{Na}(t) = \mu_{Nah} C \frac{W*d}{L} * (f(V_{mem}, t) - E_{Na} - V_{Nah})^2, \left(\frac{(V_{Nam} - V_{Nah})}{2} < f(V_{mem}) < V_{Nah} \right) \quad (6)$$

where $f(V_{mem}, t)$ is the output of the amplifier circuit^{2,20,21}. These equations relate the inherent capacitances (C), mobilities (μ), thresholds (V), and device dimensions (W , d , L) of each channel's device to its corresponding output current, enabling OFF-ON behavior in the K channel and OFF-ON-OFF behavior in the Na channel^{2,20,21}. Thus, the behaviors defined by the Hodgkin-Huxley equations are replicated. Lastly, the leakage channel is simply replicated by the device leakage.

However, since this is not novel, we have included a simplified explanation and a referral to the work done by Harikesh et al. for an in-depth explanation. These lines are as follows:

"As previously discussed in Harikesh et al., the intrinsic properties of the anti-ambipolar OECT enable replication of neuron spiking activity by mimicking the key activation and inactivation parameters of the Na and K channels presented in the classical Hodgkin-Huxley equations^{2,20,21}. The bilayer anti-ambipolar OECTs can replicate the relationship between these gating variables and the conductance of the channel due to the OFF-ON-OFF behavior of the device's transfer curve, and the OFF-ON behavior of the Gate-Drain connected voltage sweep (Figure 4). Specifically, the K channel behavior can be replicated with the traditional linear regime OECT equations where the activation gating variable is akin to the current dependance on applied and threshold voltage^{2,20,21}. On the other hand, the Na channel behavior can be replicated as an accumulation mode and depletion mode material if VGS is before or after the peak, respectively. This then replicates the activation and inactivation gating variable based on the changes in mobility and threshold voltage when switching from OFF to ON and ON to OFF, respectively^{2,20,21}. For a detailed explanation of the relationship between device equations and the Hodgkin-Huxley equations, we refer the reader to Supplementary Notes 3 and 4 within Harikesh et al^{2,20,21}."

2. Compared with recent memristor/transistor-based HH neuron models (e.g., "Third-order nanocircuit elements for neuromorphic engineering," Nature, 2020; "Ion-tunable antiambipolarity in mixed ion–electron conducting polymers enables biorealistic organic electrochemical neurons," Nature Materials, 2023), what advantages do vOECT devices offer as neuromorphic devices in terms of device footprint and circuit complexity?

We refer the reviewer to SI Table 1, where the metrics cited in the comment are shown. Using our bilayer vOECTs, the single device footprint is reduced by nearly half of that presented in the Nature Materials paper referenced. At the same time, our design approach enables to control the spikes threshold across a larger range, which is of computational interest as mentioned in the main text. Comparing our circuit metrics to memristor technology, our system allows primarily to reduce circuit complexity, as fewer capacitors and resistors are required. Moreover, it naturally displays (thanks to the OMIECs employed in its design) an ion-based modulation mechanisms, which closely mimic the biological neurons dynamics. In terms of biomimicry, this system enables spiking amplitudes closer to the physiological range (from 1200-400mV to 300-100mV), and lower constant power consumption by roughly 2 orders of magnitude. Furthermore, the high capacitance of OECT devices enables opportunities to drive spiking circuits without external capacitors thereby significantly reducing circuit complexity even further as demonstrated in "Ion-tunable antiambipolarity in mixed ion–electron conducting polymers enables biorealistic organic electrochemical neurons," Nature Materials, 2023".

3. In Section 4, the authors highlight some retina-related functions but do not emphasize that "the mutual preprocessing of wavelength and light intensity detection signals from rod and cone cells follows an associative rule." They have constructed various vOECT-based circuits to realize logical functions, yet these do not fully capture the remarkable capabilities of visual processing. The importance of logical functions in visual information processing needs to be more strongly emphasized in the paper.

We thank the reviewer for highlighting this point of confusion and enabling us to improve the manuscript. We have included a line within the main text to more strongly emphasize the importance of logical functions in the visual information processing systems, which reads as follows:

"Encoding via logic functions enables many of the algorithms used in the retina to be implemented in neuromorphic hardware, as the retinal preprocessing does not only rely on spike encoded signal, but also on graded potentials computing^{14,30,38–41}."

We feel this explanatory sentence captures the need of being able to encode information using logic functions for retinal applications in a direct manner, which are beyond the spike encoding. In addition, expanded systems were modeled showing multiple horizontal cells (ie logic functions) working together to enable increasing complex functions such as ON-Center OFF-Surround behaviors, within SI Note 4. As such, to expand upon this and address another comment we have included lines within SI Note 4 on the optimization of bilayer characteristics to produce more bio-mimetic functions and thus increased computation. These lines read as follows:

"In all cases, the system was able to encode light intensity into firing intensity within the ON window of the BBL/PEDOT bilayer devices. To closely replicate the retinal bandpass and intensity encoding, a bilayer vOECT could be constructed exhibiting an asymmetric triangular transfer curve, where a linear increase in drain current with gate voltage can provide intensity encoding across a larger dynamic range, while a sharp (nearly instantaneous) switch towards the second OFF state can preserve the bandpass capabilities of the bilayer device. As such, additional material optimization would be needed to combine a p-type with high current and transconductance near threshold with an n-type exhibiting a near linear trend in drain current with gate voltage."

4. What about the stability of these vOECTs? This information would be crucial for bioinspired neuromorphic device applications.

We thank Reviewer 3 for their insightful questions. We discussed the stability of these devices within SI Note 1, Figure S5, and Section 2, where the devices are stable up to 750 cycles both when tested immediately and when tested after 3 months of storage in ambient conditions. Furthermore, we discuss possible mechanisms of degradation, including single device degradation and swelling-induced increases in top contact resistance, as well as routes for improving the stability of the bilayer devices via improved p-type materials and usage of flexible top electrodes.

5. The tunable threshold characteristic of the device is intriguing. The authors should discuss in detail the reasons for this tunable threshold and its impact on neuronal circuit control within the paper.

We thank the Reviewer for their interest in the benefits of threshold control in various applications such as the neuronal circuit control and appreciate the opportunity to clarify this benefits which are clearly enabled by the presented bilayer approach.

The tunable threshold enables the design of reconfigurable logic gates operating on a range of voltage ranges, enabling optimization of logic functions as well as the opportunity to overlay multiple logic circuits together to achieve more neuromorphic functions. The tunability has also been demonstrated to enable gaussian probabilistic neural networks, as demonstrated by Sebastian et al. *Nat. Commun.* 2019 and discussed in the Introduction section of this work. As for the impact of tunable thresholds on spiking neurons and its benefits, while briefly discussed in the Introduction and Conclusion sections, those sections have been updated to clarify the impact of the manuscript which now better highlights the benefits of tunable neural thresholds:

"For example, neurons with different action potential thresholds have been identified in the brain. These threshold variations represent a fundamental mechanism enabling neural microcircuits sharing a similar structure to perform a variety of computational tasks, as well as enabling increased robustness, synchronicity, sensitivity, and dynamic range in the whole central nervous system^{17–20}."

"On the other hand, controlling vOECT characteristics allows for the fabrication of various neurons with differing threshold voltages, thereby mimicking the diversity of internal neuron characteristics present in the human brain and enabling a route towards application of traditional neural microcircuits for a variety of computational tasks, such as sensory pathways and motor control^{17–20}."

These updates are supported by four new citations: Gast, R., Solla, S. A. & Kennedy, A. Neural heterogeneity controls computations in spiking neural networks. *Proc. Natl. Acad. Sci.* 121, e2311885121 (2024), Gjorgjieva, J., Meister, M. & Sompolinsky, H. Functional diversity among sensory neurons from efficient coding principles. *PLoS Comput. Biol.* 15, e1007476 (2019)., Tsunozaki, M. & Bautista, D. M. Mammalian somatosensory mechanotransduction. *Sens. Syst.* 19, 362–369 (2009), Hodson-Tole, E. F. & Wakeling, J. M. Motor unit recruitment for dynamic tasks: current understanding and future directions. *J. Comp. Physiol. B* 179, 57–66 (2009), which describe the computational power of neural threshold diversity as well as its applications in mammalian systems.

6. What is the significance of mimicking retinal pathways? The article should demonstrate specific applications (e.g., bio-interfaced closed-loop electronics).

The primary goal of mimicking a retinal pathway with our system was to demonstrate the spatial and computational benefits of using bilayer vOECTs as a base to design and build more complex neuromorphic circuits. Correspondingly, we highlight the potential for future applications where these devices can help organic based retinal implants to reach increasingly pre-processing capabilities resembling that of the human visual system. To address this within the main text, we included a line within the conclusion:

"This system can serve as a building block for the future integration of the various computational components of the retina thereby enabling bio-interfaced, closed loop, bidirectional retinal prosthetics. These systems rely on the combination of spike encoding processed by spiking circuits (retinal ganglion cells) and graded potentials processed by logic circuits (horizontal cells), thus they not only overcome the biocompatibility and sensing limitations of current technologies but also promote a intrinsic neuromorphic computation paradigm through advanced anti-ambipolar devices^{1,3,5,30,38-41.}",

as well as two additional references (Wu, K. Y., Mina, M., Sahyoun, J.-Y., Kalevar, A. & Tran, S. D. Retinal Prostheses: Engineering and Clinical Perspectives for Vision Restoration. *Sensors* 23, (2023)., Matrone, G. M. et al. A modular organic neuromorphic spiking circuit for retina-inspired sensory coding and neurotransmitter-mediated neural pathways. *Nat. Commun.* 15, 2868 (2024)) that review retinal prosthetics and demonstrate additional OECT retinal pathways, respectively.

Reviewer #4 (Remarks to the Author):

I co-reviewed this manuscript with one of the reviewers who provided the listed reports. This is part of the Nature Communications initiative to facilitate training in peer review and to provide appropriate recognition for Early Career Researchers who co-review manuscripts

REVIEWERS' COMMENTS

Reviewer #1 (Remarks to the Author):

All comments have been addressed.

Reviewer #2 (Remarks to the Author):

In the revised version of the manuscript the authors addressed all my comments. I believe that the manuscript is suitable for publication.

Reviewer #3 (Remarks to the Author):

The authors have addressed my comments and revised/improved the manuscript. In my opinion it can be accepted for publication.

Reviewer #4 (Remarks to the Author):
